# On the diurnal, weekly, seasonal cycles and annual trends in atmospheric $CO_2$ at Mount Zugspitze, Germany during 1981–2016

Ye Yuan[1], Ludwig Ries[2], Hannes Petermeier[3], Thomas Trickl[4], Michael Leuchner[1,5], Cédric Couret[2], Ralf Sohmer[2], Frank Meinhardt[6], Annette Menzel[1,7]

[1]Department of Ecology and Ecosystem Management, Technical University of Munich (TUM), Freising, Germany
[2]German Environment Agency (UBA), Zugspitze, Germany
[3]Department of Mathematics, Technical University of Munich (TUM), Garching, Germany
[4]Institute of Meteorology and Climate Research, Atmospheric Environmental Research (IMK-IFU), Karlsruhe Institute of Technology (KIT), Garmisch-Partenkirchen, Germany
[5]Springer Nature B.V., Dordrecht, Netherlands
[6]German Environment Agency (UBA), Schauinsland, Germany
[7]Institute for Advanced Study, Technical University of Munich (TUM), Garching, Germany

*Correspondence to*: Ye Yuan (yuan@wzw.tum.de)

**Abstract.** A continuous, 36-year measurement composite of atmospheric carbon dioxide ($CO_2$) at three measurement locations of Mount Zugspitze, Germany was studied. For a comprehensive site characterization of Mount Zugspitze, analyses of $CO_2$ weekly periodicity and diurnal cycle were performed to provide evidence for local sources and sinks, showing clear weekday to weekend differences with dominantly higher $CO_2$ levels during the daytime of the weekdays. A case study of atmospheric trace gases (CO and NO) and passenger numbers to the summit indicate that closeby $CO_2$ sources did not result from tourist activities but obviously from anthropogenic pollution in the near vicinity. Such analysis of local effects is an indispensable requirement for selecting representative data at orographic complex measurement sites. The $CO_2$ trend and seasonality were then analyzed by background data selection and decomposition of the long-term time series into trend and seasonal components. The mean $CO_2$ annual growth rate over the 36-year period at Zugspitze is $1.8 \pm 0.4$ ppm yr$^{-1}$, which is in good agreement with Mauna Loa station and global means. The peak-to-trough difference of the mean $CO_2$ seasonal cycle is $12.4 \pm 0.6$ ppm at Mount Zugspitze (after data selection: $10.5 \pm 0.5$ ppm), which is much lower than at nearby measurement sites at Mount Wank ($15.9 \pm 1.5$ ppm) and Schauinsland ($15.9 \pm 1.0$ ppm), but following a similar seasonal pattern.

## 1. Introduction

Long-term records of atmospheric carbon dioxide ($CO_2$) improve our understanding of the global carbon cycle, as well as long- and short-term changes, especially at remote background locations. The longest continuous measurements of atmospheric $CO_2$ started in 1958 at Mauna Loa, Hawaii, initiated by investigators of the Scripps Institution of Oceanography (Pales and Keeling, 1965). The measurements were performed on the north slope of the Mauna Loa volcano at an elevation of 3397 m above sea level (a.s.l.), thus at long distances from $CO_2$ sources and sinks. Later, additional measurement sites

were established for background studies of global atmospheric $CO_2$, such as the South Pole (Keeling et al., 1976), Cape Grim, Australia (Beardsmore and Pearman, 1987), Mace Head, Ireland (Bousquet et al., 1996), and Baring Head, New Zealand (Stephens et al., 2013). Along with sites located in Antarctica or along coastal/island regions, continental mountain stations offer excellent options to observe background atmospheric levels due to high elevations that are less affected by local influences, for example, Mount Waliguan, China (Zhang et al., 2013), Mount Cimone, Italy (Ciattaglia, 1983), Jungfraujoch, Switzerland, and Puy de Dôme, France (Sturm et al., 2005).

Although mountainous sites experience less impact from local pollution and represent an improved approach to background conditions compared with stations at lower elevations, we cannot fully dismiss the influence of local to regional emissions. This influence largely depends on air-mass transport and mixing within the moving boundary layer height. LIDAR measurements show that air from the boundary layer is orographically lifted to approximately 1–1.5 km above typical summit heights during daytime in the warm season (Carnuth and Trickl, 2000; Carnuth et al., 2002). A 14-year record of atmospheric $CO_2$ at Mount Waliguan (3816 m a.s.l.), China, reveals significant diurnal cycles and depleted $CO_2$ levels during summer that are mainly driven by biological and local influences from adjacent regions, although the magnitude and contribution of these influences are smaller than those at other continental or urban sites (Zhang et al., 2013). At the Mt. Bachelor Observatory (2763 m a.s.l.), U.S.A., atmospheric $CO_2$ variations were studied in the free troposphere and boundary layer separately, where wildfire emissions were observed to drive $CO_2$ enhancement at times (McClure et al., 2016). However, it still remains unclear to exactly what extent elevated mountain sites are influenced by local activities and how to characterize better local sources and sinks at such stations. It is difficult to make quantitative conclusions on the anthropogenic and biogenic contributions to these measurements (Le Quéré et al., 2009). Analyzing weekly periodicity may be a potential indicator since periodicity represents anthropogenic activity patterns during one week (seven days) without the influence of natural causes (Cerveny and Coakley, 2002). From the prespective of modeling and satellite observational system, studies have shown that the weekly variability has implications on the quantification and verification of anthropogenic $CO_2$ emissions, as well as diurnal variability (e.g., Nassar et al., 2013; Liu et al., 2017). Regarding in-situ measurements, results from Ueyama and Ando (2016) clearly indicate the presence of elevated weekday $CO_2$ emissions compared with weekend and/or holiday $CO_2$ emissions at two urban sites in Sakai, Japan. Cerveny and Coakley (2002) detected significantly lower $CO_2$ concentrations on weekends than on weekdays at Mauna Loa, which was assumed to result from anthropogenic emissions from Hawaii and nearby sources.

In this study, we present a composite 36-year record of atmospheric $CO_2$ measurements (1981–2016) at Mount Zugspitze, Germany (2962 m a.s.l.). The objective of this study is to achieve an improved measurement site characterization with respect to historical $CO_2$ data in terms of diurnal and weekly cycles, and to produce a consistent overall analysis of $CO_2$ trend and seasonality. The $CO_2$ measurements were performed at three locations on Mount Zugspitze: at a pedestrian tunnel (ZPT), at the summit (ZUG), and at the Schneefernerhaus (ZSF) on the southern face of the mountain. In addition, $CO_2$ measurements were taken at the nearby lower mountain station, Wank Peak (WNK), but for a shorter time period. Short-term variations of weekly $CO_2$ periodicities and diurnal cycles were evaluated for Mount Zugspitze. In addition, a case study

combing atmospheric CO and NO measurements and records of passenger numbers was used to examine weekday-weekend influences. Then the results for the $CO_2$ annual growth rates and seasonal amplitudes were studied separately via trend-seasonal decomposition and compared with $CO_2$ data for the comparable time period (1981–2016) at the GAW Regional Observatory Schauinsland, Germany (SSL) and the GAW Global Observatory Mauna Loa, Hawaii (MLO), as well as the

global $CO_2$ means calculated by the NOAA/ESRL and the World Data Centre for Greenhouse Gases (WDCGG).

## 2. Experimental methods and data

### 2.1. Measurement locations

Mount Zugspitze is located approximately 90 km southwest of Munich, Germany. The nearest major town is Garmisch-Partenkirchen (GAP, 708 m a.s.l.). Measurements of $CO_2$ were first performed between 1981 and 1997at a southward-facing

balcony in a pedestrian tunnel (ZPT, 47°25′ N, 10°59′ E, 2710 m a.s.l) situated about 250 m below the summit of Mount Zugspitze, which joined the ancient summit station of the first Austrian cable car  to the Schneefernerhaus (Reiter et al., 1986).  The Schneefernerhaus was a hotel until 1992 when it was rebuilt into an environmental research station. From 1995 until 2001, a new set of measurements were made at a sheltered laboratory on the terrace of the summit (ZUG, 47°25′ N, 10°59′ E, 2962 m a.s.l.). These two measurement periods were performed by the Fraunhofer-Institute for Atmospheric

Environment Research (IMK-IFU), and, since 1995 these measurements have been carried out on behalf of the German Environmental Agency (UBA). Since 2001, to continue contributing to the GAW Programme, $CO_2$ measurements have been performed at the Environmental Research Station Schneefernerhaus (ZSF, 47°25′ N, 10°59′ E, 2656 m a.s.l.). Approximately 100 m below the Schneefernerhaus, the glacier plateau Zugspitzplatt can be reached from the valley via cable cars or cogwheel trains. The Zugspitzplatt descends eastward via a moderate to steep slope across the Knorrhütte towards the

Reintalangerhütte as shown in Fig. 1 (Gantner et al., 2003).

### 2.2. Instrumental setup and data processing

$CO_2$ mole fractions were processed separately because of different measurement locations and time periods at Mount Zugspitze as described above. Information on the first time period (ZPT) was collected based on personal communication with corresponding staff, logbooks, and literature research (Reiter et al., 1986). The $CO_2$ measurement at ZPT was

continuously performed with different, consecutively used instrument models (i.e., the URAS-2, 2T, and 3G) of nondispersive infrared (NDIR) technique. The measured values were corrected by simultaneously measured air pressure with a hermetically sealed nitrogen-filled gas cuvette due to no flowing reference gas used. Two commercially available working standards (310 and 380 ppm of $CO_2$ in $N_2$) were used for calibration every day at different hours. The $CO_2$ concentration in this gas bottle was compared in short intervals with a reference standard provided by UBA which was adjusted to the

Keeling standard reference scale.

At ZUG the sampling line consisted of a stainless steel tube with an inner core of borosilicate glass and a cylindrical stainless steel top cup against intake of precipitation. The inlet with the structure of a small mast ended approximately 4 m on the top of the laboratory building, which is situated on the Zugspitze summit platform (see Fig. 1b). Inside the laboratory a turbine with a fast real-time fine control ensured a constant sample inflow of 500 l/min of in-situ air. The borosilicate glass

tube (about 10 cm diameter) continued inside the laboratory, providing a number of outlets from where the instruments could get the sample air for their own analyses. The measurement and calibration were performed with a URAS-3G device and an Ansyco mixing box. The mixing controller allowed automatic switching for up to four calibration gases and sampling air by a self-written calibration routine using Testpoint software. The linear two-point calibration enveloping the actual ambient values with low and high $CO_2$ concentrations was taken at every $25^{th}$ hour. Every six months the working standards were

checked and re-adjusted, when required, to the standard reference scale by inter-comparison measurements with the station standards.

At ZSF the same construction principle was applied for atmospheric sampling. There, the mast ends about 2.5 m above the pavement of the research terrace at the $5^{th}$ floor in an altitude of 2670 m a.s.l. Measurements of $CO_2$ at Schneefernerhaus continued thereafter to the present with a modified HP 6890 by using gas chromatography (GC) with an intermediate

upgrade in 2008 (Bader, 2001; Hammer et al., 2008; Müller, 2009). In 2012 and 2013, because of an instrumental failure of the GC, $CO_2$ data were recorded with a cavity ring-down spectrometer (CRDS, Picarro EnviroSense 3000i) connected to the same air inlet, which had been installed in parallel since 2011. The GC calibrations were carried out at 15 minute intervals using working standards (near-ambient), which had been calibrated with station standards from the GAW Central Calibration Laboratory (CCL) operated by the NOAA/ESRL Global Monitoring Division. The GC data acquisition system (see

Supplementary Fig. S1) produced a calibration value every 15 minutes and two values from the sampled air based on one chromatogram every five minutes. Calibration factors and metadata were used to convert raw data into the final data product. Invalid and unrepresentative data due to local influences were flagged according to a pollution list. The measurement quality was controlled by comparison with simultaneous measurements of identical gas (CRDS) or with measurements of other trace substances and meteorological data, and additional support from station logbooks and checklists. The data were flagged

according to quality control results. In principle, the acquisition system stores all measured data (flagged or not) and never discards them. Drifts in the working standards were controlled by a quasi-continuously measured second target and a regular two-month inter-comparison between the working standard and NOAA station standards, performing corrections as needed. Calibration for CRDS was performed automatically with three different concentrations every 12 hours. Until 2013 the calibrations were performed automatically every 24 hours with one concentration, very close to the ambient value. Every

two months the concentrations were re-checked to the station reference standards.

Additional atmospheric $CO_2$ measurements throughout the GAP area were performed between 1978 and 1996 at Mount Wank summit (WNK, 47°31′ N, 11°09′ E, 1780 m a.s.l.) using a URAS-2T instrument. The Wank Observatory is located in an alpine grassland just above the tree line (Reiter et al., 1986; Slemr and Scheel, 1998). Detailed information on the $CO_2$ measurements at Schauinsland (SSL, 47°55′ N, 7°54′ E, 1205 m a.s.l.) and Mauna Loa, Hawaii (MLO, 19°28′ N, 155°35′ W,

3397 m a.s.l.), which we use to compare the results of this study with, can be found in Schmidt et al. (2003) for SSL and Thoning et al. (1989) for MLO. The $CO_2$ data from these measurement sites and from Mount Zugspitze locations were considered as validated data set (Level 2: calibrated, screened, artefacts and outliers removed), without any further data processing prior to the selection of representative data.

**2.3. Offset adjustment**

According to NOAA CMDL (http://ds.data.jma.go.jp/wcc/co2/co2_scale.html), no significant offsets are documented between the calibration scales WMO X74 and WMO X85 and the current WMO mole fraction scale. However, for the three-year parallel $CO_2$ measurements at ZPT and ZUG (1995–1997), clear offsets of –5.8 ± 0.4 ppm ($CO_{2, ZPT}$ minus $CO_{2, ZUG}$, $1 \cdot SD$) were observed. The major reason for this bias is assumed to be the pressure-broadening effect in the used gas

analyzers and the different gas mixtures used in the standards, $CO_2/N_2$ vs. $CO_2/air$, the so called "carrier gas correction (CGC)" (Bischof, 1975; Pearman and Garratt, 1975). It is known from previous studies that the measured $CO_2$ concentration, when using $CO_2/N_2$ mixtures as reference, is usually underestimated by several ppms for the URAS instruments, and such offsets vary from different types of analyzers (Pearman, 1977; Manning and Pohl; 1986). The carrier gas effect varies even between the same type of analyzer as well as with replacement of parts of the analyzer (Griffith et al., 1982; Kirk Thoning,

personal communication, August 1, 2018). Due to lack of information and impossible on-site experiments with previous calibration standards, an offset adjustment to the $CO_2$ data set at ZPT was made for further analyses based on the offsets in data computed in the overlapping years instead of a physically derived correction. A single correction factor

$$G = 0.956 + 0.00017 \cdot C_{ZPT} \tag{1}$$

was applied to the ZPT data while $C_{ZPT}$ denotes the $CO_2$ concentrations at ZPT. Because of the same calibration mixtures, an additional adjustment was applied to the $CO_2$ concentrations at WNK by calculating the $CO_2$ differences between ZPT and

WNK. A detailed description on the offset adjustment of CGC with potential errors is given in the supplement. Two similar CGCs by Manning and Pohl (1986) at Baring Head, New Zealand and Cundari et al. (1990) at Mt. Cimone, Italy, were comparable in magnitude to our offset adjustment.

On the other hand, there were 9 consecutive months, from April to December 2001, of parallel atmospheric $CO_2$ measurements at both ZUG and ZSF, based on which an inter-comparison between the two series was made. The offset

between these two records attained an average of 0.1 ± 0.4 ppm ($CO_{2, ZUG}$ minus $CO_{2, ZSF}$, $1 \cdot SD$), which fulfills the requirement of the GAW Data Quality Objective (DQO, ±0.1 ppm) for atmospheric $CO_2$ in the Northern Hemisphere. Therefore, no adjustments regarding this offset were applied to the data sets.

In this study, we took $CO_2$ measurements during the corresponding time intervals at ZPT (1981–1994), ZUG (1995–2001), and ZSF (2002–2016) to assemble a composite time series for Mount Zugspitze over 36 years. Nevertheless, we always treat

measurements from each location separately for further analyses. At WNK, as well as at SSL and MLO, we used measured $CO_2$ data starting from 1981 for time consistency with measurements at Mount Zugspitze.

## 2.4. ADVS data selection

Adaptive Diurnal minimum Variation Selection (ADVS), a recently published, novel statistical data selection strategy, was used to ensure that the data were clean and consistent with respect to the state of a locally unaffected lower free troposphere at the measurement sites (Yuan et al., 2018). ADVS, which was originally designed to characterize mountainous sites,
selects data based on diurnal patterns with the aim of selecting optimal data that can be considered representative of the lower free troposphere. To achieve this, variations in the mean diurnal $CO_2$ were first evaluated and a time window was selected based on minimal data variability around midnight, at which point data selection began. The data outside the starting time window were examined on a daily basis both forward and backward in time for the day under consideration, by applying an adaptive threshold criterion. The selected data results represent background $CO_2$ levels at the different
measurement sites.

ADVS data selection was applied to all $CO_2$ records based on the same threshold parameters, followed by examining the starting time window and calculating the percentages of the ADVS-selected data. Figure 2(a) shows the $CO_2$ time series before and after ADVS data selection. We also evaluated the starting time windows resulting from ADVS data selection with the detrended mean diurnal cycles as described in Yuan et al. (2018) for each measurement site in Fig. 2(b). The number of
ADVS-selected data is summarized as percentage per hour in the total number of all $CO_2$ data in Fig. 3. A detailed description and discussion is given in Sect. 3.1.

## 2.5. Mean symmetrized residual

Weekly periodicity was calculated using the "Mean Symmetrized Residual" (MSR) method, which was originally applied to atmospheric $CO_2$ data (Cerveny and Coakley, 2002). The MSR method focuses on variations in mean values by the days of
the week. Daily deviations from the seven-day (consecutive) averages are calculated to account for the most likely emission cycles. Then, the MSR values are derived by averaging the differences for each single day. Additionally, only the MSR values with no data gaps in all the seven differences are considered as valid. Finally, all the MSR values are aggregated into overall mean values for each day of the week. In addition, the MSR values are standardized so that the sum of all the seven values is equal to 0 (Cerveny and Coakley, 2002).

## 2.6. STL decomposition

The Seasonal-Trend Decomposition technique (STL) was applied to decompose the $CO_2$ time series into trend, seasonal and remainder components individually (Cleveland et al., 1983; Cleveland et al., 1990), which, in previous studies, has been a commonly applied method (e.g., Stephens et al., 2013; Hernández-Paniagua et al., 2015). Locally weighted polynomial regressions were iteratively fitted to all monthly values in both an outer and an inner loop. According to Cleveland et al.
(1990) and Pickers and Manning (2015), we set the trend and seasonal smoothing parameters to 25 and 5, respectively. The

CO$_2$ time series at each site / location were aggregated into monthly averages and, then, decomposed by STL. Missing monthly values were substituted by spline interpolation.

To study the trend and seasonality, we firstly intended to apply STL decomposition to the ADVS-selected time series. However, due to multiple occurrences of consecutively missing values in the ADVS-selected monthly averages, especially for measurement sites at lower elevations (WNK and SSL), it was more practical to use the original CO$_2$ time series without ADVS data selection for STL decomposition, to preserve time series continuity (Pickers and Manning, 2015). There is one missing six-month time interval at ZUG in 1998 (July to December). Thus STL was performed separately for the time periods before (1995.01–1998.06) and after (1999.01–2001.12) the gap. Nevertheless, we still applied STL decomposition to the ADVS-selected data sets from Mount Zugspitze and Mauna Loa, since these selected time series were applicable. Only at ZPT, due to greatly missing data at the beginning (1981 and 1982) of the ADVS-selected data set, we only used the ADVS-selected results starting from 1983. Individual figures of each STL-decomposed component at all stations can be found in the supplement.

For annual growth rates we did not include the WNK time series due to shorter time periods of available data. Monthly trend components were first aggregated into annual mean values. Then, the annual CO$_2$ growth rates were calculated as the difference between the CO$_2$ value of the current year and the value from the previous year (Jones and Cox, 2005). The mean seasonal cycle was aggregated directly from the monthly seasonal components by month. To observe potential deviations on the regional and global scale, we compared the trend and seasonality derived from the STL decomposed components respectively at Zugspitze with other measurement sites. We included the globally averaged marine surface monthly mean data from NOAA (www.esrl.noaa.gov/gmd/ccgg/trends/) and data for the global mean mole fractions from WDCGG (WMO, 2017) as references, and processed these data based on the identical STL decomposition routine. All the statistical analyses described above (including ADVS, MSR, and STL) were performed in the R environment (R Core Team, 2018).

## 3. Results and discussion

### 3.1. ADVS selection and diurnal variation

The resulting ADVS-selected CO$_2$ data showed a clear linkage of the percentage of selected data and the altitude of the measurement site. Among the continental stations, the percentage increased with altitude. Lower percentage indicates higher data variability due to lower elevation and proximity to local sources and sinks. At Schauinsland, the percentage of CO$_2$ data by the ADVS selection was 6.3% while the percentages at Mount Zugspitze reached 9.9% (ZPT), 19.5% (ZUG), and 13.6% (ZSF), respectively. A moderate percentage of 6.3% was also derived at Mount Wank. However, regarding the elevated mountain station Mauna Loa on the island of Hawaii, a much higher percentage (40.0%) of CO$_2$ data was selected by ADVS as representative of its background concentration mainly due to the very limited nearby anthropogenic sources as well as mostly clean, well-mixed air arriving there. A similar result for an island mountain station can be found in Yuan et al. (2018) where a percentage of 36.2% was computed for the CO$_2$ measurements at the station Izaña on Tenerife Island (28º19′ N,

16º30′ E, 2373 m a.s.l.). This can also be explained by the detrended mean diurnal cycles shown in Fig. 2(b) and Fig. 3. The mean diurnal cycle at MLO only exhibits a clear trough during daytime, especially starting from 12:00h local time (LT), which is believed to be influenced by the vegetation activity (photosynthesis) in the surroundings. The same effect can be seen at WNK and SSL, but with larger magnitudes and earlier occurrences of the minima because of their lower locations

closer to $CO_2$ sinks. In contrast, at these two sites the $CO_2$ maxima in the diurnal cycles were not as clearly noticeable as at Mount Zugspitze due to anthropogenic sources and high biogenic respiration. At the three locations of Mount Zugspitze, the $CO_2$ peaks in the mean diurnal cycles are driven by the late-morning convective upslope wind, which was relatively obvious at both ZUG and ZSF. However, from the perspective of data selection, a significantly higher percentage of $CO_2$ data was selected at ZSF compared with ZPT although there is only a small difference in altitude of around only 70 m. This proves

that ZSF is capable to capture more background conditions than ZPT during the day. Nevertheless, based on the starting time window computed for ADVS selection, we found that, in general, most stations exhibited similar starting time windows beginning around midnight and the ADVS data selection was applied systematically by including more data around these hours (see Fig. 3), which confirmed our assumption of background conditions during midnight for the ADVS data selection (Yuan et al., 2018).

**3.2. Weekly periodicity**

For a better characterization of the differences among the measurement locations at Mount Zugspitze, the mean $CO_2$ weekly cycles were analyzed as a function of mean MSR values (see Fig. 4a). The mean MSR values at the MLO for the corresponding time intervals were also calculated. Most weekly cycles exhibited no clear peaks or patterns for both sites. However, the magnitude of MSR data variability is mostly higher at Zugspitze with a maximum on Thursdays. The only

significant weekday-weekend difference is observed at ZSF in terms of the 95% confidence interval, which shows weekly maxima and weekly minima on Thursday and Saturday, respectively (peak-to-trough difference: 0.76 ppm). Gilge et al. (2010) observed similar phenomena when studying $O_3$ and $NO_2$ concentrations at Alpine mountain stations including Zugspitze. Clear weekly cycles, with enhanced $O_3$ levels on working days, were observed at ZSF in summer, with weekly maxima and minima on Thursday and Sunday, respectively. For $NO_2$, maximum mixing ratios on working days and

minimum ratios on Sundays at neighboring stations were observed, generally suggesting an anthropogenic impact at all elevations.

We obtained more insights into the weekly $CO_2$ cycle at Mount Zugspitze by comparing the mean diurnal cycles of weekdays and weekends (see Fig. 4b). Detrended mean diurnal cycles at ZSF, from Sunday to Saturday, were calculated by subtracting the daily averages from the daily data between 2002 and 2016. In the morning around 9 to 10 a.m. LT the $CO_2$

levels at ZSF are higher on weekdays than weekends, while $CO_2$ diurnal patterns during the rest of the week are relatively stable. Such weekly cycles are not observable at ZPT and ZUG nor at WNK and SSL (see Supplementary Fig. S18). At ZPT, there are less variations in the diurnal cycle compared to ZSF, indicating that this location does not receive the effect of regular local anthropogenic working activities and hence it is more representative of lower free tropospheric conditions

regarding this aspect. The weekday-weekend differences at ZSF are possibly due to local working patterns, whereas the absence of this pattern at lower sites may indicate influences from a more regional reservoir. In fact, ZSF is closed on the weekends and, thus, influenced by less immediate anthropogenic activities.

### 3.3. Case study on atmospheric CO, NO, and passenger numbers at Zugspitze

To study further the potential sources and sinks for such weekday-weekend differences in the $CO_2$ diurnal cycles at ZSF, we analyzed atmospheric CO and NO data at ZSF and the daily, combined number of cable car and train passengers to Zugspitzplatt and to the Zugspitze Summit in 2016. Atmospheric CO and NO are known to be good indicators of local anthropogenic influences due to highly variable short-term signals and are thus helpful to identify potential $CO_2$ sources (Tsutsumi et al., 2006; Sirignano et al., 2010; Wang et al., 2010; Liu et al., 2016). In this study, we used atmospheric NO due

to its short lifetime based on rapid atmospheric $NO_2$ formation with resulting altitude-dependent $O_3$ surplus, indicating the presence of sources at closer distances. The CO and NO data shown in Fig. 5 include data that was flagged during data processing, because for the delivery to GAW World Data Centers the logged and recognized work dependent concentration peaks are flagged. A clear weekday-weekend difference is observed for both CO and NO. Only weekdays are characterized by multiple short-term atmospheric CO events and higher atmospheric NO peaks during the daytime (mostly around 9 a.m.

LT), which fits perfectly with daytime peaks in $CO_2$ diurnal cycles. A general fluctuating pattern in NO throughout the week is thought to originate from heating of the Zugspitzplatt and changing work with combustion engines. On the other hand, the daily number of passengers at Zugspitze (see Fig. 5c) shows a clear weekday–weekend pattern with higher number of passengers on the weekends. However, increased numbers of passengers on the weekends do not correspond to higher levels of CO and $CO_2$, indicating that measured $CO_2$ levels are not significantly influenced by tourist activities nearby. Instead, it is

more likely that anthropogenic working activities are the main driver of weekly periodicity.

### 3.4. Trend

Based on the STL decomposed results, the mean annual growth rate of the 36-year composite record at Mount Zugspitze from the three measurement locations is $1.8 \pm 0.4$ ppm yr$^{-1}$, which is consistent with the SSL ($1.8 \pm 0.4$ ppm yr$^{-1}$), MLO ($1.8 \pm 0.2$ ppm yr$^{-1}$), and global means (NOAA: $1.8 \pm 0.2$ ppm yr$^{-1}$; WDCGG: $1.8 \pm 0.2$ ppm yr$^{-1}$). The mean annual growth rates

from the ADVS-selected data sets at Mount Zugspitze and Mauna Loa also result in the identical value of 1.8 ppm yr$^{-1}$. Then, we divided the entire time period (1981–2016) into three time blocks corresponding to the different locations at Mount Zugspitze in order to observe potential differences with respect to other sites separately (see Table 1). The results show good agreement of each location of Mount Zugspitze with other measurement sites (also for the ADVS-selected results) as well as a clearly increasing trend of the annual growth rates over these three time blocks. Only the mean annual growth rate between

1995 and 2001 at ZUG is obviously lower than at the other sites. This can be explained by the missing monthly values in 1998 and thus in turn the annual growth rates of 1998 and 1999 were left out for the average. However, the annual growth

rates of these two years reached anomalous peaks at most sites (see details later in Sect. 3.6). Möller (2017) also mentioned that 1981 to 1992 growth rates at both German stations and MLO were identical.

### 3.5 Seasonality

For the overall seasonality, Figure 6 presents the mean seasonal cycles for the STL decomposed seasonal components. We
observed similar patterns in the SSL and WNK seasonal cycles, with mean peak-to-trough differences of 15.9 ± 1.0 and 15.9 ± 1.5 ppm, respectively. The composite data set at Mount Zugspitze results in a lower amplitude (12.4 ± 0.6 ppm), but still exhibits a similar seasonality influenced by active biogenic processes (mainly photosynthesis) in summer compared with SSL and WNK (Dettinger and Ghil, 1998). As vegetation grows with rising temperatures (approaching summer), $CO_2$ levels decrease due to more and more intense photosynthetic activities till a minimum in August. In addition, with rising
temperatures, locally influenced air masses reach Mount Zugspitze more often due to "alpine pumping" (Carnuth et al., 2002; Winkler et al., 2006). As such, air sampled in summer is more frequently mixed with air from lower levels, which is characterized by lower $CO_2$ concentrations, intensifying the August minimum. Anthropogenic activities and plant respiration dominate the increases in concentration in the winter (January to April). This influence appears to be stronger at SSL and WNK than at Mount Zugspitze. Lower levels of $CO_2$ and a one-month delay, from February to March, of the seasonal
maximum at Mount Zugspitze are in agreement with the expectation of thermally driven orographic processes that drive the upward transport of $CO_2$ from local sources, as well as limited human access to Mount Zugspitze and the prevailing absence of biogenic activities at such high elevations. Regarding the resulting seasonal cycles based on ADVS-selected Zugspitze data sets, similar patterns were observed but with a lower amplitude (10.5 ± 0.5 ppm) as well as a two-month shift of the seasonal maximum to April.

The Mauna Loa $CO_2$ record is characterized by a seasonal maximum in May and a minimum in September with a peak-to-trough difference of 6.8 ± 0.1 ppm, which agrees with observations from Dettinger and Ghil (1998) and Lintner et al. (2006). The ADVS-selected results for MLO also show a similar pattern with a lower amplitude of 6.6 ± 0.1 ppm. Global means exhibited the lowest seasonal amplitudes, 4.4 ± 0.1 ppm (NOAA) and 4.8 ± 0.0 ppm (WDCGG). Compared with WDCGG, NOAA global mean fits better the seasonal cycle of MLO supporting the presence of a typical Marine Boundary Layer
(MBL) condition for the levels of background $CO_2$ in the atmosphere. On the other hand, the WDCGG global mean includes continental characteristics for its calculation, thus exhibiting a slightly more continental signature which can be equally seen in the seasonal cycles at continental sites, such as Mount Zugspitze. April and October appear to be the important months that indicate the switch of either $CO_2$ source to sinks or vice versa for the continent.

We then examine in more detail the seasonal cycles at ZPT, ZUG, and ZSF. Despite the close proximity, there are
differences in their seasonal amplitudes (ZPT: 11.9 ± 1.2 ppm; ZUG: 11.2 ± 1.0 ppm; ZSF: 13.3 ± 0.7 ppm). Good agreement is shown between $CO_2$ seasonal cycles from April to June and from October to December. However, significantly higher levels of $CO_2$ were evident at ZSF from January to March as well as lower levels from July to September. After data selection with lower seasonal amplitudes of 10.3 ± 1.3 ppm (ZPT_ADVS), 10.3 ± 1.2 ppm (ZUG_ADVS), and 10.9 ± 0.6

ppm (ZSF_ADVS), similar differences of the $CO_2$ levels in the seasonal cycles could be observed. These results indicate that factors such as elevation and measurement surroundings strongly determine the air-mass composition via local vertical transport. The amount of air-mass transport via orographic lifting affects the three locations differently. The lower elevation station, ZSF, apparently captures more mixed air masses due to a daytime up-valley flow along the Reintal (Gantner et al., 2003) as well as a slightly southeastern flow from the Inntal (see Fig. 1) that is less frequent for the higher locations (ZPT or ZUG). In addition, comparably postponed seasonal maxima at ZUG and ZPT from March to April show delayed onset of convective upwind air-mass transport and changing Planetary Boundary Layer (PBL) compositions.

### 3.6. Inter-annual variation

To study the inter-annual variability, we focused on the percentages of ADVS selection, the growth rates, and the seasonal amplitudes. The annual percentages from ADVS data selection are shown for years without missing monthly averages (see Fig. 7a). An exceptionally high percentage at Zugspitze in 2000 resulted from careful and intensive filtering of the original $CO_2$ data. The total number of validated 30-min data points in 2000 is 4634, while the amount of data for other years ranges from 8754 to 15339 (except for 1998, with only 6-month data, the total number of 30-min $CO_2$ data is 6441). As described in the previous section, the annual growth rates are plotted in Fig. 7b. The annual $CO_2$ seasonal amplitudes are calculated as the difference between the yearly maximum and minimum monthly $CO_2$ values from the STL decomposed seasonal components (see Fig. 7c).

Focusing on the annual percentages from ADVS-selected representative data after 1990, we calculated the mean annual percentages at Mount Zugspitze locations, for the time periods between 1990 and 2001 (2000 was not included for ZUG), and 2002 and 2016. We observe significantly higher percentages at ZPT and ZUG ($18.5 \pm 2.4\%$) than at ZSF ($13.6 \pm 1.1\%$) at a 95% confidence interval. These percentages are different from SSL ($4.2 \pm 0.5$ vs. $4.2 \pm 0.6\%$) and MLO ($43.5 \pm 1.4$ vs. $42.1 \pm 1.6\%$). A likely explanation is that there are systematically different air-mass transport characteristics reaching each of these locations. Higher percentages at ZPT and ZUG indicate that these locations are capable of capturing more air masses that have traveled over long distances along the mountains. These air masses trap air that ascends from many Alpine valleys, but also from remote source regions up to intercontinental scale (Trickl et al., 2003; Huntrieser et al., 2005). On the other hand, ZSF is dominated by mixing air masses that have traveled along the Zugspitzplatt area, which contain higher levels of $CO_2$ due to daily, local anthropogenic sources during winter and convective upwind during seasons without snow cover that are characterized by lower concentrations of $CO_2$ at lower altitudes. Such patterns in the data are also evident in the annual growth rates and seasonal amplitudes. The overall patterns at Mount Zugspitze agree with SSL and WNK. However, SSL and WNK exhibit more variation in the annual growth rates and higher seasonal amplitude levels (see Fig. 7b and 7c). In addition, slightly higher seasonal amplitudes for the WDCGG global mean compared with the NOAA one can be explained by the WDCGG global mean calculation method, which includes more continental stations (WMO, 2017).

Anomalies in the annual growth rates are frequently observed, which are possibly explained by climatic influences such as the El Nino-Southern Oscillation (ENSO), volcanic activity, and extreme weather conditions (Keeling et al., 1995; Jones and

Cox, 2001; Francey et al., 2010; Keenan et al., 2016). One of the largest positive annual growth rate anomalies occurred in 1998 and is clearly seen in all the records (aside from ZUG with missing values), which is attributed to a strong El Niño event (Watanabe et al., 2000; Jones and Cox, 2005). Similar signals are found in 1988, especially at MLO and in global means. Such anomalies are more clearly observed in the global and seaside time series. Regarding continental sites, inter-

annual signals may be hidden by more intense land influences rather than global effects. Moreover, positive consecutive anomalies between 2002 and 2003 are clearly observed at ZSF and SSL, which are potentially due to anomalous climatic conditions, such as the dry European summer in 2003 that led to an increasing number of forest fires. These events are also observable in the MLO and global means but at smaller scale (Jones and Cox, 2005). At all German sites, clear negative anomalies, due to violent eruptions of the El Chichón and Mt. Pinatubo volcanoes and the subsequent volcanic induced

surface cooling effect are observed after stratospheric aerosol maxima above Garmisch-Partenkirchen in 1983 and 1992, respectively (Lucht et al., 2002; Frölicher et al., 2011; Frölicher et al., 2013; Trickl et al., 2013). This effect is only slightly visible in the MLO and global means despite the fact that volcanic aerosol spread over the entire globe.

However, the reasons for some anomalies are still unclear. These include the negative anomalies during 1985 and 1986 at all Germans sites. Certain anomalies in the annual percentages and seasonal amplitudes also derive from extremely low ADVS

selection percentages beginning at 1984 and continuing until 1990, with peaks in seasonal amplitudes between 1985 and 1986. This is the reason why we calculated the mean annual ADVS selection percentage beginning at 1990. We assume that local influences mask similar physical mechanisms at the sites. However, annual percentages at the MLO also have similar characteristics. Therefore, it is still unclear what triggers such distinct inter-annual data variability across measurement sites. Another clear negative annual growth rate anomaly occurred in 2014 across all sites. Such anomalies still require further

investigation, but are beyond the scope of this study.

## 4. Conclusions

In this study, we presented a time series analysis of a 36-year $CO_2$ measurement record at Mount Zugspitze in Germany together with a thorough study of the weekly periodicity combined with diurnal cycles. Even though it is challenging to quantify local sources and sinks, this study shows that it is possible to gain information on variation in this regard. Compared

with the GAW Regional Observatories at Schauinsland and Wank Peak, as well as the GAW Global Observatory at Mauna Loa, Mount Zugspitze proves to be a highly suitable site for monitoring background levels of air components using proper data selection procedures. The long-term trend at Zugspitze agrees well with that at Mauna Loa and global means. The seasonality and short-term variations show similar patterns, but are considerably less influenced by local to regional mechanisms than the lower elevation stations at Schauinsland and Wank Peak. Inter-annual variations also correlate well

with anomalous global events. However, several anomalies still exist across most stations that lack clear explanations. These anomalies require further investigation possibly by analyzing correlations between extreme events and historical meteorological or hydrological data. Finally, we conclude that, at Zugspitze, we cannot neglect local to regional influences.

Regarding the seasonal amplitude, Mount Zugspitze is significantly more influenced by biogenic activity, mostly in the summer compared with Mauna Loa and global means. On the other hand, weekly periodicity analysis provides a clear picture of local $CO_2$ sources that potentially result from human working activities, especially at ZSF. Overall, this study provides detailed insights into long-term atmospheric $CO_2$ measurements, as well as site characteristics at Mount Zugspitze.

We propose the application of this type of analysis as a systematic tool for the physical and quantitative classification of stations with respect to their lower free tropospheric representativeness. As an additional component in this analysis, weekly periodicity can be used to analyze anthropogenic influences. The systematic application of this approach to larger continental or global regions can serve as a basis for more quantitative analyses of global greenhouse gases trends such as $CO_2$. Based on the physical foundation of the methodology presented here, we suggest that these techniques can be applied to other

greenhouse gases such as $SF_6$, $CH_4$, and aerosols.

**Acknowledgements**

This study was supported by a scholarship from the China Scholarship Council (CSC) under Grant CSC No. 201508080110. Our thanks go to the support from a MICMoR Fellowship through the KIT/IMK-IFU to Ye Yuan. Our thanks go to Gourav Misra for the geographical map of the measurement locations. Our thanks go to James Butler and Kirk Thoning from NOAA

for their indispensable discussions on the problematic nature of representing and comparing data on different older and actual $CO_2$ scales. The $CO_2$, CO and NO measurements at Zugspitze Schneefernerhaus, Platform Zugspitze of the GAW Global Observatory Zugspitze/Hohenpeissenberg, and $CO_2$ measurements at Schauinsland are supported by the German Environment Agency (UBA). The IMK-IFU provided data from the Zugspitze tunnel and summit. Our thanks go to Dr. H. E. Scheel from the IMK-IFU for his high quality data measurement until 2001 at the Zugspitze Summit (ZUG). During a long

period, Dr. Scheel, who passed away in 2013, led the in-situ measurement program at the Zugspitze summit with a high level of expertise and diligence. Former IFU staff members helped us to reconstruct details of the measurements. We would also like to thank the operating team at the Environmental Research Station Schneefernerhaus for supporting our scientific activities and to the Bavarian Ministry for Environment for supporting this High Altitude Research Station. Finally, our gratitude goes to the Bavarian Zugspitze Railway Company for the passenger data in 2016.

**Data availability**

NOAA global mean: ftp://aftp.cmdl.noaa.gov/products/trends/co2/co2_mm_gl.txt.
WDCGG global mean: https://ds.data.jma.go.jp/gmd/wdcgg/pub/global/2017/co2_monthly_20171030.csv.
$CO_2$ records (also including CO and NO) of all GAW Observatories which were used in this study can be found from the World Data Centre for Greenhouse Gases (WDCGG): https://ds.data.jma.go.jp/gmd/wdcgg/wdcgg.html.

The daily passenger number data for Zugspitze were provided by Bayerische Zugspitzbahn.

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

Table 1: Mean annual $CO_2$ growth rates in ppm $yr^{-1}$ at the 0.95 confidence interval based on three time blocks for all measurement sites / locations studied (SSL – Schauinsland; WNK – Mount Wank; ZPT – pedestrian tunnel at Mount Zugspitze; ZUG – Zugspitze summit; ZSF – Zugspitze Schneefernerhaus; MLO – Mauna Loa; WDCGG and NOAA – global means). ADVS means the data were selected by ADVS method. This comparison refers to data from all years including the corresponding time period for all stations. Measurement sites or locations where data are not available for calculating the corresponding time blocks are shown as "–".

| Time block | SSL | WNK | ZPT | ZPT _ADVS | ZUG | ZUG _ADVS | ZSF | ZSF _ADVS | MLO | MLO _ADVS | WDCGG | NOAA |
|---|---|---|---|---|---|---|---|---|---|---|---|---|
| 1981–1994 | 1.5 ± 0.5 | 1.4 ± 1.1 | 1.5 ± 0.8 | 1.5 ± 1.4 | – | – | – | – | 1.4 ± 0.3 | 1.4 ± 0.3 | 1.4 ± 0.4 | 1.4 ± 0.3 |
| 1995–2001 | 1.7 ± 1.1 | – | – | – | 1.3 ± 0.8 | 1.5 ± 0.5 | – | – | 1.8 ± 0.5 | 1.8 ± 0.5 | 1.8 ± 0.4 | 1.7 ± 0.5 |
| 2002–2016 | 2.2 ± 0.7 | – | – | – | – | – | 2.2 ± 0.4 | 2.2 ± 0.4 | 2.2 ± 0.2 | 2.2 ± 0.2 | 2.2 ± 0.2 | 2.2 ± 0.2 |

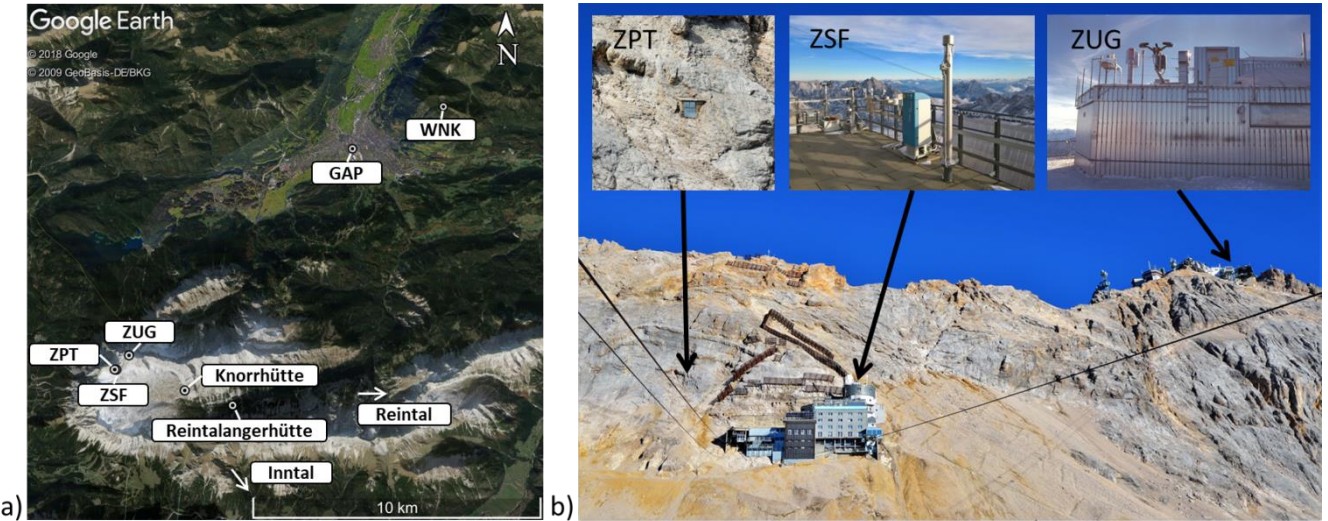

**Figure 1: (a) Map showing the study area (GAP – Garmisch-Partenkirchen; WNK – Mount Wank; ZPT – pedestrian tunnel at Mount Zugspitze; ZUG – Zugspitze summit; ZSF – Zugspitze Schneefernerhaus). (b) A photograph showing the locations (ZPT, ZSF, and ZUG) at Mount Zugspitze where atmospheric $CO_2$ measurements were performed.**

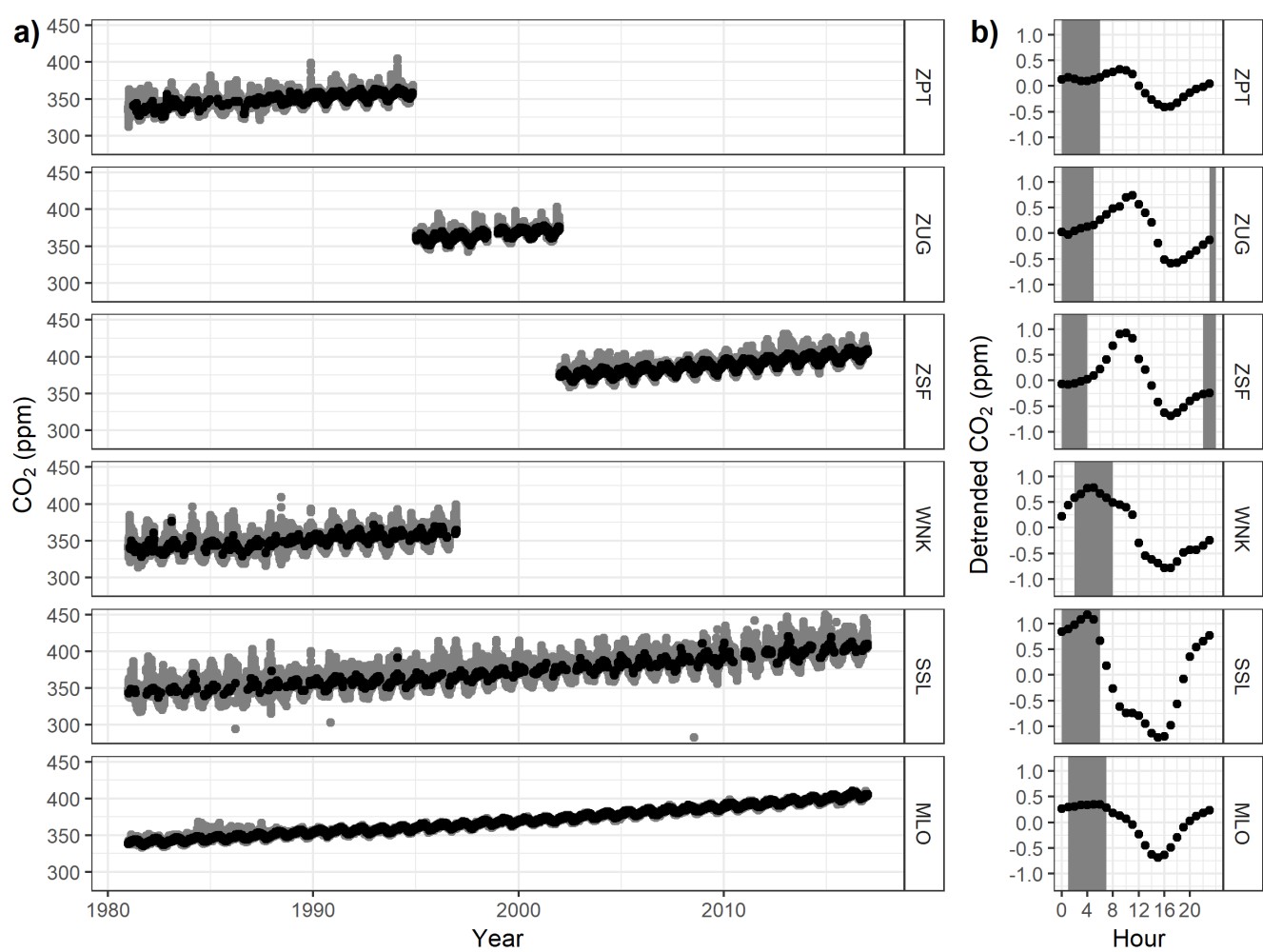

**Figure 2: a) Time series plot of 30-min averaged CO$_2$ concentrations measured at Mount Zugspitze (ZPT, ZUG, and ZSF) and Wank (WNK), and hourly averaged CO$_2$ concentrations measured at Schauinsland (SSL) and Mauna Loa (MLO) with ADVS-selected results. b) Detrended mean diurnal cycles with starting time windows (in grey) for ADVS data selection.**

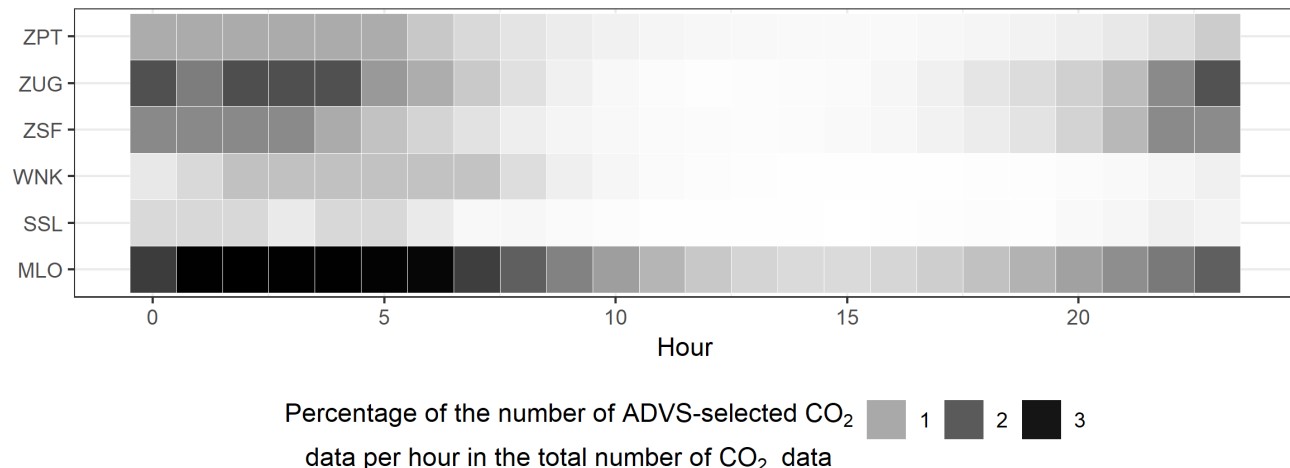

**Figure 3: Frequency of the percentages of the number of ADVS-selected CO$_2$ data for each hour (0 to 23) in the total number of CO$_2$ data. In the shown greyscale grey means 1%, 2% and black means 3% of the data.**

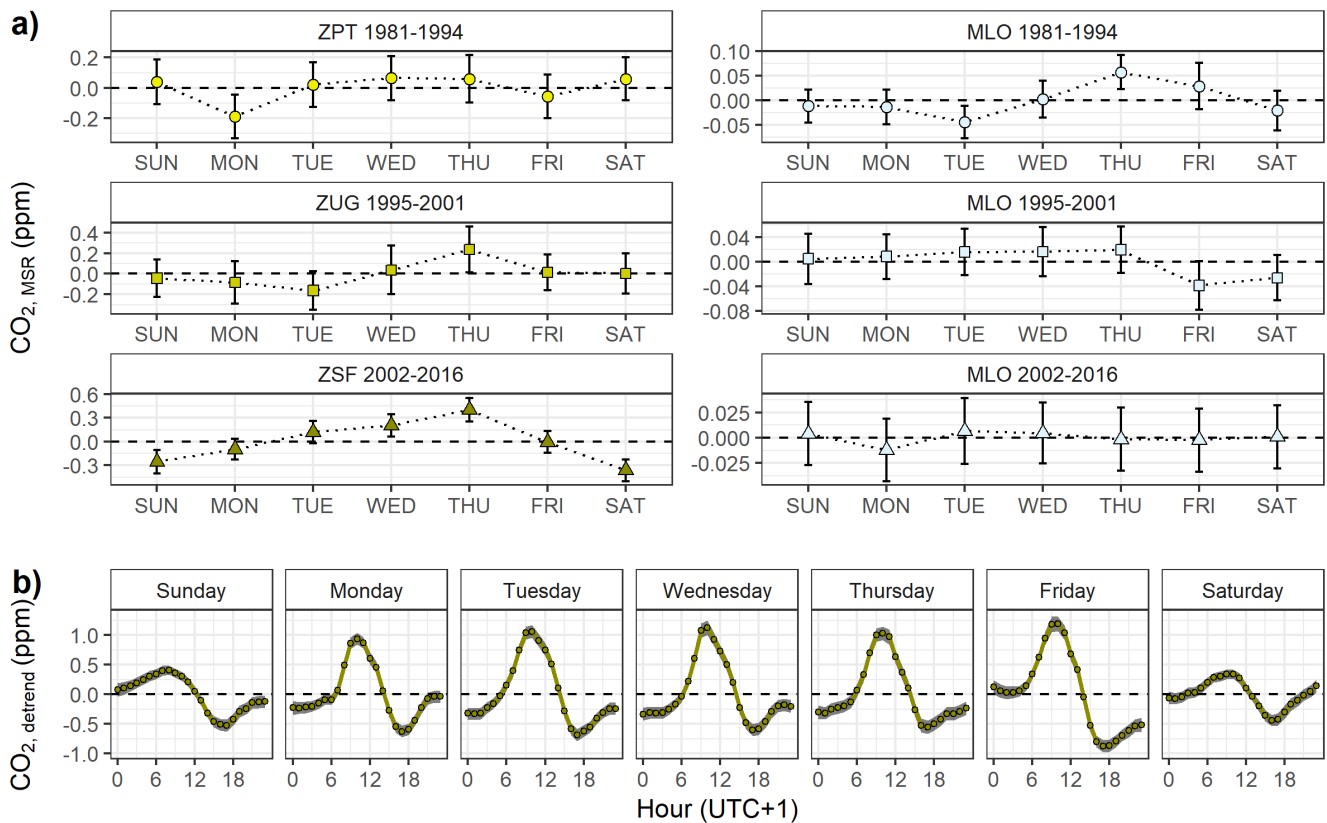

**Figure 4: a) Mean MSR CO$_2$ values at Mount Zugspitze and MLO as a function of the weekday. Mean MSR values are adjusted such that they sum to 0. b) Detrended mean CO$_2$ diurnal cycles at ZSF by weekday from 2002 to 2016. Uncertainties at a 95% confidence interval are shown by the shaded areas.**

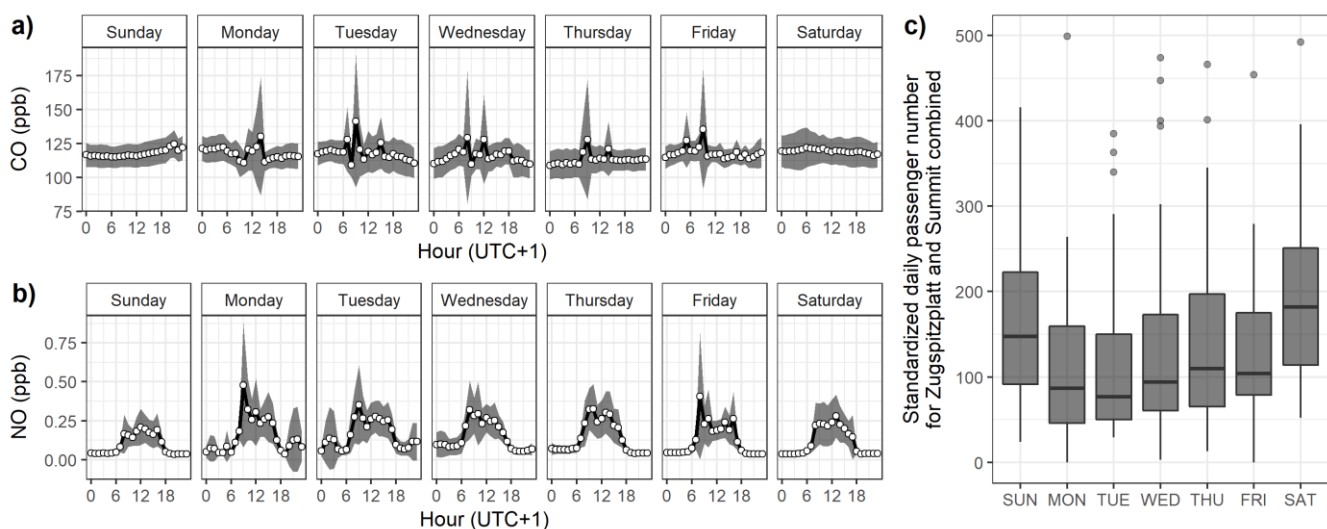

Figure 5: Mean diurnal plots at ZSF during 2016 by weekday for a) CO, b) NO, and c) the standardized daily passenger number at the Zugspitzplatt and Zugspitze summit combined.

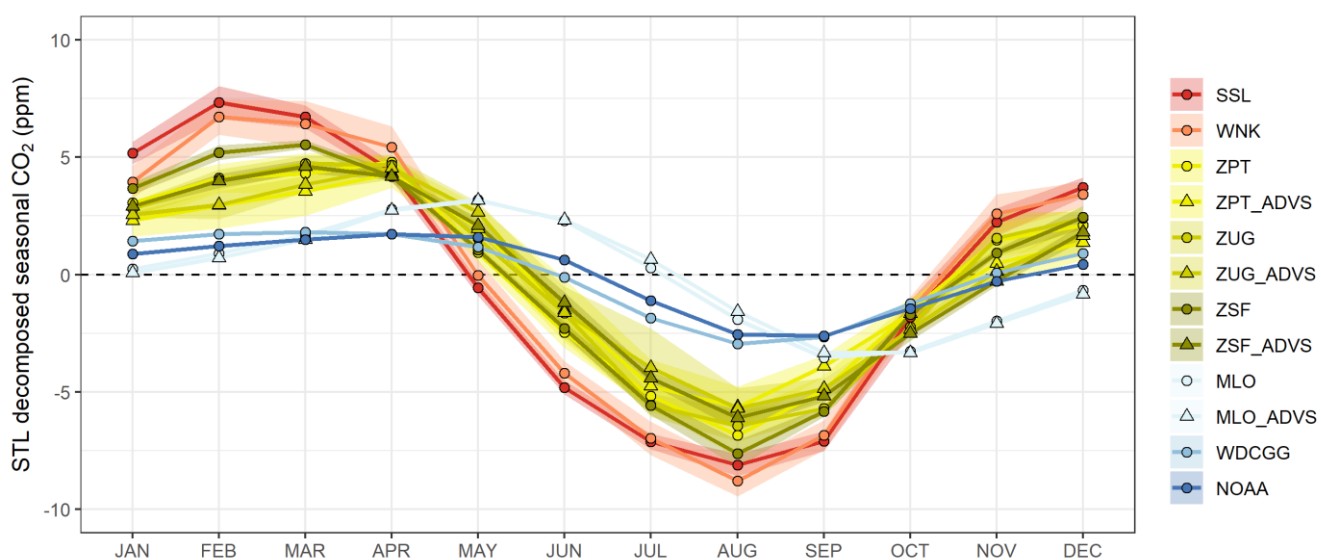

**Figure 6: Mean CO$_2$ seasonal cycles from the STL seasonal component at each measurement site or location. Uncertainties at a 95% confidence interval are shown by the shaded areas with corresponding color.**

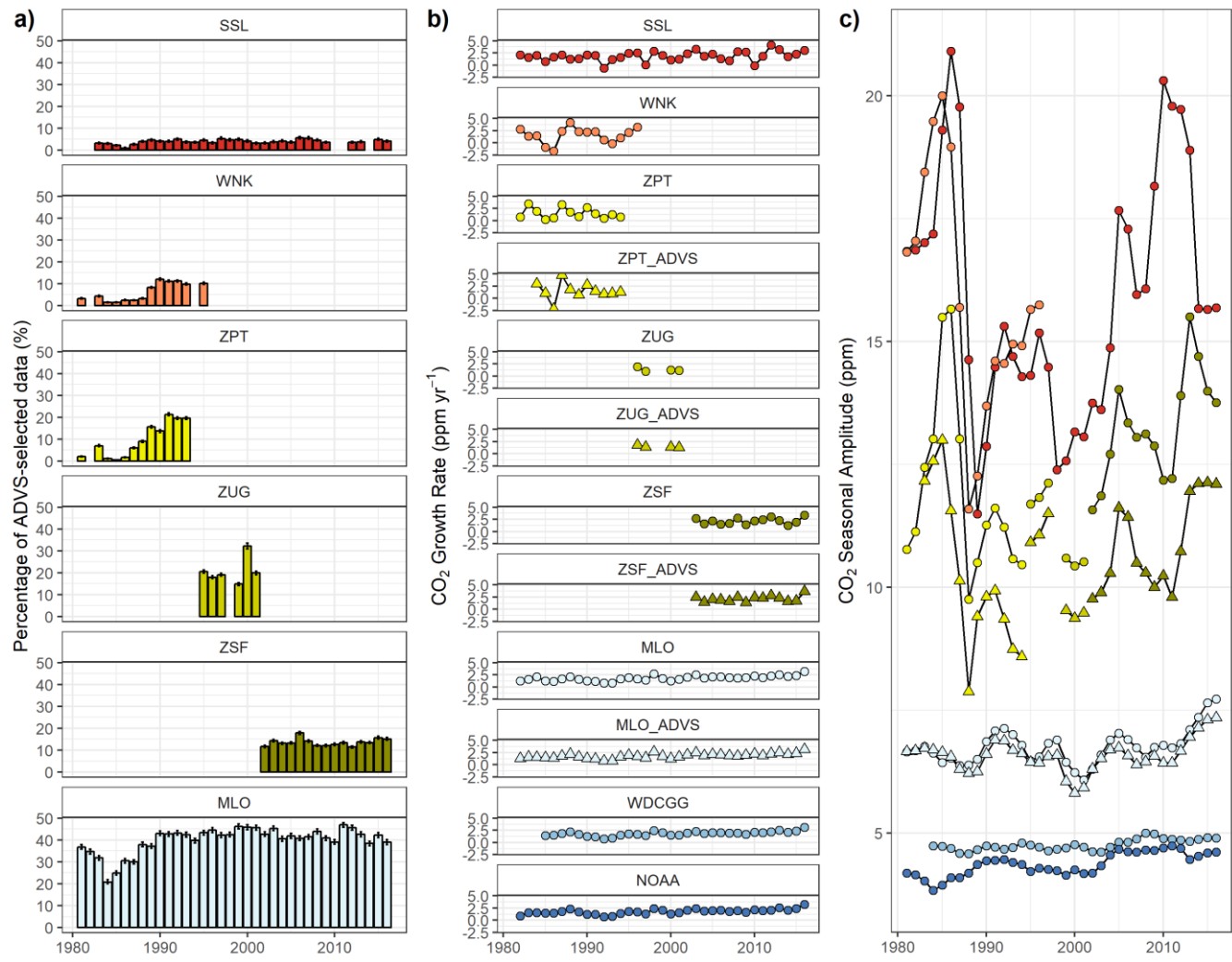

**Figure 7: a) Annual ADVS-selected percentages. b) Annual CO$_2$ growth rates and global means from the NOAA and the WDCGG. The calculated growth rates are shown at the beginning of the year. Since the time period starts in 1981, the values of growth rates start in 1982. WDCGG data is only available starting 1984. c) Annual CO$_2$ seasonal amplitudes.**