# Peer review of "On the diurnal, weekly, seasonal cycles and annual trends in atmospheric CO2 at Mount Zugspitze, Germany during 1981–2016"

_Atmospheric Chemistry and Physics, 2018_

## Short Comment (SC1) · 11 Sep 2018

This is an interesting manuscript. It is good to see that diurnal, weekly and seasonal cycles in CO2 are being investigated by the ground-based CO2 measurement community. The importance of the seasonal cycle is obvious. The value of accounting for the diurnal and weekly cycles has been pointed out in previous work from a model perspective such as Nassar et al. (2013) and Liu et al. (2017), which would be worthwhile to cite in revisions to this manuscript. These cycles in anthropogenic CO2 emissions have implications for the design of future satellite systems aiming to quantify anthropogenic CO2 emissions at various scales to support emission reduction efforts.

[Figure]

**References**

R. Nassar, L. Napier–Linton, K.R. Gurney, R.J. Andres, T. Oda, F.R. Vogel, F. Deng. Improving the temporal and spatial distribution of CO2 emissions from global fossil fuel emission datasets, Journal of Geophysical Research: Atmospheres, 118, 917-933, doi:10.1029/2012JD018196, 2013. https://agupubs.onlinelibrary.wiley.com/doi/10.1029/2012JD018196

Y. Liu, N. Gruber, D. Brunner. Spatiotemporal patterns of the fossil-fuel CO2 signal in central Europe: results from a high-resolution atmospheric transport model. Atmos. Chem. Phys., 17, 14145–14169, 2017, https://doi.org/10.5194/acp-17-14145-2017

---

## Referee Comment (RC1) · Anonymous Referee #1 · 22 Oct 2018

Review of the manuscript: "On the diurnal, weekly, seasonal cycles and annual trends in atmospheric CO2 at Mount Zugspitze, Germany during 1981–2016" by Y.Yuan et al. (Atmos. Chem. Phys. Discuss., https://doi.org/10.5194/acp-2018-850)

The paper is describing the long term CO2 monitoring program at Zugpitze, Germany. Actually the time series is a composite from three periods during which the sampling location, the method and instrument were different: ZPT (1981-1997), ZUG (1995-2001), and ZSF (2001-ongoing). Consequently a major issue to be addressed in this study is the consistency of the three datasets, in order to determine if they can be grouped in a single series and with what limitations. I think this part is not detailed

enough. The three datasets are merged for analysis of different time scale variabilities, although several indicators show that they differ significantly. A scientist using the Zugspitze long term time series without consideration of the change in the sampling location could misinterpret the signal. For this reason I would recommend the authors to clarify the uncertainties associated to such a merging of the different dataset.

2.1. Measurement sites: I would suggest not using the term 'sites' to distinguish between the three sampling locations (ZPT, SUG, ZSF) at Zugspitze. It is a source of confusion here and there in the manuscript.

Schneefernerhaus (ZPT, 47°25′ N, 10°59′ E, slightly below the summit): please give the elevation asl

'Information for the first and second time periods were mainly collected based on personal communication with corresponding staff and logbooks': at least it would be good to get information on the general setup of the system (dryer, calibration, and data selection . . .).

Do you use the data already selected (according to time of the day or other criteria) from previous site managers ?

2.3. Offset adjustment: The offset between the two sites (ZPT and ZUG) is huge with a large dispersion (5 to 6 ppm). A more detailed analysis of this offset, looking at its variation in time (and instrumental change), as a function of the atmospheric pressure, or CO2 concentration must be provided.

I support the hypothesis that the carrier gas effect can explain most of the differences between ZPT and ZUG, since it is well known that the CO2 concentration in air when using N2 mixtures as references, is under-estimated by few ppm. However, a discussion on this issue must be provided by the authors, with references to previous studies based on similar NDIR instruments (e.g. Pearman et al., Tellus, 1975; Griffith et al., Tellus, 1982). Is the observed differences compatible with what we can expect considering the literature, and the atmospheric pressure at this altitude site ?

I understand that you have not applied the offset correction (-0.11ppm) between ZUG and ZSF. Please make it clear.

2.4. ADVS data selection: "The percentages of ADVS-selected data are ... 13.5% for Zugspitze": have you merged all three Zugspitze stations together in this analysis ? Does it mean there is none significant differences between them ?

Could you provide some statistics of the hours which are selected at Zugspitze as representative of the background according to ADVS method ?

2.5. STL decomposition Missing monthly values were substituted by spline interpolation: do you allow an interpolation of large data gaps like several consecutive months ?

"...especially for measurement sites at lower elevations": I am confused about which sites you are referring to. Low altitude sampling locations at Zugspitze, or other sites like Schauinsland ? Can you be more specific about the data gaps at stations, since it would make much more sense to use background data (after ADVS selection) for the seasonal and trend analysis, especially when comparing at other large scale time series.

3.1. Trend and seasonality "Only the mean annual growth rate between 1995 and 2001 at the ZUG site is much lower than the other sites due to missing values in 1998": Not clear for me why the 6 months data gap in summer 1998 decreases so much the total trend over the period 1995 to 2001. Please clarify.

"Amplitudes of 15.44 and 14.89 ppm": For most signals I would suggest rounding the values to one decimal place.

The comparison of the seasonal cycles would be much more meaningful with background selected data. By the way do you use also all data (without selection) at SSL, WNK and MLO sites, or do you use the data selected by the station's managers at

those sites ? Please clarify. It could be interesting to see if the ZSF site remains more influenced by the air from the valleys, compared to ZPT and ZUG, once you have selected the nighttime values at all sites.

"there are slight differences in seasonal amplitudes (ZPT: 10.86 ppm; ZUG: 11.14 ppm; ZSF: 13.09 ppm) among the three sites": I would not call a 2 ppm signal a slight difference ! A major signal to look at for such long term time series in North Hemisphere would be a possible trend in the amplitude of the seasonal cycle which could indicate a trend in the way the biosphere is interacting with atmospheric $CO_2$. Graven et al., 2013 described for example increasing trends of the seasonal $CO_2$ amplitude of 0.32 % per year at Mauna Loa and 0.60 % per year at Point Barrow. Considering a mean amplitude of about 12 ppm you could expect a trend of 1.4 to 2.5 ppm over the 36 years period of measurement at Zugspitze (assuming the MLO and BRW trends).

Figure 3: the significant differences you show on figure 3b with the 3 sampling locations should prevent you from mixing those three dataset together as you do in figure 3a.

3.2. Inter-annual variations Abnormal high percentage at Zugspitze in 2000: I do not understand the sentence on line 5/6 suggesting that a careful and intensive selection was performed in 2000. Is the selection process different from the other years ?

Again, due to the differences between the three sampling locations (especially ZSF which is more influenced by air uplifted from the valleys) I think you should differentiate them in figure 4.

3.3. Weekly periodicity I would suggest to discuss short-term variabilities (weeks and daily) before trend and inter-annual variations.

I do not see the interest of comparing the weekly variations at Zugspitze to the one observed at Mauna Loa.

---

## Referee Comment (RC2) · Anonymous Referee #2 · 8 Nov 2018

General Comments

This paper outlines a set of CO2 data records collected over >30 years at locations within the German alps, specifically the methods used and trends observed. These long-term continental records, although more complicated to interpret than coastal records, are important. As such details of these records, like those given in this paper, should be published and the records themselves made publicly available. Unfortunately, there is a distinct lack of detail when it comes to the calibration approach used, in particular for the older data records. This needs to be rectified before publication. The paper also has a number of sentences which are confusing to read and would

benefit greatly from the Copernicus copy editing service or the help of a native English speaker. I have attempted to note these in the technical corrections section and offered some suggestions for how they could be clarified. I feel that only with the addition of significant detail in relation to the calibration approach and a revision of the language used should the paper should be published.

Specific Comments

Abstract

Examining weekend-weekday variability in order to comment on which fluxes are driving $CO_2$ signals is a powerful tool. This, along with the outcomes of such a study should be highlighted in the abstract. At the moment the reference to it is rather vague, "indicating potential $CO_2$ sources", and could easily be strengthened.

2.1 Measurement sites

There needs to be a description of the sampling method. Was there a small mast at these locations with an intake cup or were the instruments just measuring the air around them?

2.2 Data processing

If you're presenting data from the first two time periods then you need to give information on how that data (or wasn't calibrated). If they weren't calibrated then say so, and in the discussion provide an estimate of the size of the error that this will drive in the data. The GC calibration method is unclear to me. From the description it appears that you have a single working standard, the concentration of which is adjusted based on the station standards, and that this working standard is measured once every 15 mins. This will account for instrumental drift but does that mean you're assuming a linear detector response? Using GC to measure $CO_2$ is usually a more linear approach than many other $CO_2$ measurement techniques but it's not exactly linear. The effect of this non-linearity needs discussed and outlined in the text. There is no information on the

[Figure]

CRDS calibration process. If CRDS data is presented in the paper (it's not clear if it is) then this information needs to be provided.

2.3 Offset adjustment

The offset noted between ZPT and ZUG is very large – typically 6ppm – and concerning. However, it's difficult to comment on the offset adjustment used to correct for this as no information is given on how these sites are calibrated. Without further information it is impossible to know whether the offsets are driven solely by the use of $CO_2$ in $N_2$ calibration standards or other issues. It's also possible that, considering that they are different locations, that they were measuring air of different composition and part of this offset was true signal. Was any data filtering (e.g. wind speed/direction) completed prior to the comparison?

Technical and editorial corrections

The below comments are made in reference to specific areas of the text identified as page no./line no.

1/17 In this context there is no need for the definite article before "Mauna Loa" and "global means". This error occurs throughout text. For example "in good agreement with the Mauna Loa station and the global means" should read "in good agreement with Mauna Loa and global means"

1/18 It's important to include some estimate of the variability of the seasonal amplitude to give an indication of how stable it is.

1/20-22 This sentence is confusing and vague.

1/31-2/1 Please change "Apart from the sites located either in the Antarctica or along coastal/island regions, continental mountain stations also offer excellent options to observe the background atmospheric levels due to high elevations that are least unaffected..." to "Along with sites located in Antarctica or along coastal/island regions, continental mountain stations offer excellent options to observe background atmospheric levels due to high elevations that are less affected. . ."

2/2-4 This sentence is superfluous, please remove. "Presently, there are 31 Global Observatories coordinated by the Global Atmosphere Watch (GAW) network, focusing on monitoring the physical and chemical state of the atmosphere on a global scale."

2/7 Please change "lidar" to "LIDAR" it's an acronym.

2/15 Change ". . .what extend that elevated. . ." to ". . .what extent elevated. . ."

2/33 Confusing "Weekly CO2 periodicities were evaluated with the diurnal cycles for the Mount Zugspitze sites". Do you mean that the weekly periodicity was evaluated by examining diurnal cycles or that the weekly periodicity was evaluated and diurnal cycles were also evaluated? I think the former but it could be read both ways.

3/1-2 Again "In addition, we perform an atmospheric CO and NO case study to-gether with the amount of passengers at Zugspitze in 2016 as potential indicators for weekday–weekend influences." is confusing. I'm guessing you mean "A case study combining atmospheric CO and NO measurements and records of passenger numbers was used to examine weekday-weekend differences"?

3/8-11 This is confusing. Please change "The measurements were collected at a southward-facing balcony in a pedestrian tunnel (Reiter et al., 1986) from the summit of Mount Zugspitze to the Schneefernerhaus (ZPT, 47°25′ N, 10°59′ E, slightly below the summit), which was a hotel until 1992 when it was rebuilt into an environ-mental research station. From 1995 until 2001, a new set of measurements began at the summit (ZUG, 47°25′ N, 10°59′ E, 2960 m a.s.l.) at a sheltered laboratory on the terrace using a URAS-3G device." to "The measurements were collected at a southward-facing balcony of a pedestrian tunnel (Reiter et al., 1986) which joined the summit of Mount Zugspitze to the Schneefernerhaus situated Xm below the summit (ZPT, 47°25′ N, 10°59′ E). The Schneefernerhaus was a hotel until 1992 when it was rebuilt into an environmental research station. From 1995 until 2001, a new set

of measurements were made at a sheltered laboratory on the terrace of the summit (ZUG, 47°25′ N, 10°59′ E, 2960 m a.s.l.) using a URAS-3G device."

3/15-18 This section ("Zugspitzplatt, a glacier . . . shown in Fig 1. (Gantner et al., 2003)" interrupts the flow of the site descriptions. It's also unclear why it's included – I'm guessing to highlight that there are visitors nearby? Please move it to the end of the paragraph and provide more context.

3/20-21 Confusing. Were the CRDS measurements made as well as the GC measurements i.e in parallel? Or instead of due to the instrumental failure? It's unclear.

4/3 What were the concentrations of the working standards? I don't need the exact value for each cylinder but a general description would be useful. E.g. "near-ambient"

4/5-6 Confusing. The GC data acquisition system doesn't "produce" the calibration values. By their very definition acquisition systems can only acquire data. Do you mean that using the GC system chromatograms were measured every 5 minutes with the working standard measured every third chromatogram?

4/8 What is a "pollution list"?

4/8 "Simultaneous measurements of identical gas" Do you really mean that you have simultaneous measurements of CO2 made using another instrument at the same location? If so how were they made and why aren't they reported here?

4/12 If the working standard is measured every 15 minutes how often was the second target measured?

4/18 This is a really large offset, typically 6ppm. Please give the mean offset here so that readers don't have to look in the supplementary.

5/1-2 It would be useful to refer to this 36-year data record as a "compound" data record as it's actually composed of data collected at three different locations. Using this terminology would make later sections of paper clearer.

6/11 Was this done on the raw data or the ADVS filtered data?

6/23 Change "over the entire 36 year period" to "of the 36-year compound record"

6/25-26 "In general, the mean annual growth rates over the entire 36 year period at all sites agree within a range of 1.8 ppm yr–1". Which sites are you referring to here? The Zugspitze sites don't cover a 36-year period e.g. ZPT is only 16 years long. If you're referring to SSI, MLO and the global mean as referenced in the previous sentence than this sentence is redundant please remove it.

7/1-2 Please change "Möller (2017) also mentioned that growth rates at both German stations and the MLO from 1981 to 1992 were identical." To "Möller (2017) also mentioned that 1981 to 1992 growth rates at both German stations and MLO were identical."

7/8 Please change "that minimize in August" to "that reach a minimum in August".

7/10-11 Please change "Sampled air is more frequently mixed with air from lower levels, which is characterized by lower CO2 concentrations that also minimize in August." To "As such, in Summer sampled air is more frequently mixed with air from lower levels, which is characterized by lower CO2 concentrations, enhancing the August minimum."

7/17 Please change "The MLO is" to "Mauna Loa data are" or "The Mauna Loa CO2 record is"

7/18 Please change "which agree" to "which agrees"

7/18-19 Please change "Moreover, global means exhibited the lowest seasonal amplitudes of 4.33 ppm (NOAA) and 4.76 ppm (WDCGG)." To "Global means exhibited the lowest seasonal amplitudes, 4.33 ppm (NOAA) and 4.76 ppm (WDCGG)."

7/19-23 I know what you're trying to say but this section really isn't written clearly. Please correct it.

7/27-28 Please change "Apart from this, significantly higher levels of CO2 at ZSF from

January to March and lower levels from July to September cannot be neglected." To "However, significantly higher levels of CO2 are evident at ZSF from January to March and lower levels from July to September."

8/6-7 I'm confused. You state that there are an abnormally high percentage of validated data points for the year 2000 but then say there are only 4634 points but there are 15000 for the other years. Do you mean 15000 total for the remaining years or 15000 per year? If it's per year then that's seems wrong.

8/11-12 In figure 4b please colour code the sections of the compound Mt Zugspitze record for the different sites to make it easy to identify which years are ZPT, ZUG or ZSF. This would make relating this section to the figure far easier.

8/20 Please change "can also be illustrated for" to "are also evident in"

11/1-2 "Seasonal amplitude at …. compared with global sites" This sentence doesn't make sense. Please correct.

Figure 2 Please plot the data from the different sites as different colours in the bottom left hand plot to make it clear which site is being used at which time.

Figure 4 – Please add the abbreviations used in the text e.g. SSI or WNK to the titles of the plots to make comparisons between the text and the figure easier. Please colour code the sections of the compound Mt Zugspitze record for the different sites to make it easy to identify which years are ZPT, ZUG or ZSF.

Figure 5 – Match the site colour coding from figure 4 to this figure.

---

## Author Comment (AC1) · 19 Dec 2018

**Author comments on* "On the diurnal, weekly, seasonal cycles and annual trends in atmospheric CO₂ at Mount Zugspitze, Germany during 1981–2016" *by* Ye Yuan et al.**

Ye Yuan on behalf of all co-authors

Answers to **Anonymous Referee #1 (RC1)**

The referee comments are shown in black. The answers are shown in blue.

**Authors:** We would like to thank Anonymous Referee #1 for the efforts to review this manuscript and to provide very
10 helpful comments and detailed remarks. All the referee's comments have been carefully examined and addressed in the revised manuscript as well as supplement.

Review of the manuscript: "On the diurnal, weekly, seasonal cycles and annual trends in atmospheric CO2 at Mount Zugspitze, Germany during 1981–2016" by Y.Yuan et al. (Atmos. Chem. Phys. Discuss., https://doi.org/10.5194/acp-2018-
15 850)

The paper is describing the long term $CO_2$ monitoring program at Zugspitze, Germany. Actually the time series is a composite from three periods during which the sampling location, the method and instrument were different: ZPT (1981-1997), ZUG (1995-2001), and ZSF (2001-ongoing). Consequently a major issue to be addressed in this study is the consistency of the three datasets, in order to determine if they can be grouped in a single series and with what limitations. I
20 think this part is not detailed enough. The three datasets are merged for analysis of different time scale variabilities, although several indicators show that they differ significantly. A scientist using the Zugspitze long term time series without consideration of the change in the sampling location could misinterpret the signal. For this reason I would recommend the authors to clarify the uncertainties associated to such a merging of the different dataset.

**Authors**: Thank you very much for pointing this out. We have now included a detailed discussion about the offset
25 adjustment between ZPT and ZUG in the manuscript and supplement. Later on we have always made the analyses for each measurement location (ZPT, ZUG, and ZSF) separately. Throughout the manuscript we have pointed out that the results of atmospheric CO₂ measurements at Mount Zugspitze are a composite of three data sets at different locations and for different time periods, which cover an overall time length of 36 years. When using these data sets, caution is needed and it is always

recommended to discuss questions regarding specific researches with the data provider. For more detailed changes please see the following answers.

2.1. Measurement sites: I would suggest not using the term 'sites' to distinguish between the three sampling locations (ZPT, SUG, ZSF) at Zugspitze. It is a source of confusion here and there in the manuscript.

5   **Authors:** Thank you very much for the suggestion. We changed all the expressions "sites" related to ZPT, ZUG, and ZSF into "locations" throughout the manuscript.

Schneefernerhaus (ZPT, 47°25′ N, 10°59′ E, slightly below the summit): please give the elevation asl

**Authors**: Done.

'Information for the first and second time periods were mainly collected based on personal communication with
10   corresponding staff and logbooks': at least it would be good to get information on the general setup of the system (dryer, calibration, and data selection…).

**Authors**: A general instrumental setup of the measurement system at ZPT and ZUG has been implemented in the Section 2.2 (Instrumental setup and data processing).

"…The $CO_2$ measurement at ZPT was continuously performed with different, consecutively used instrument models (i.e., the
15   URAS-2, 2T, and 3G) of nondispersive infrared (NDIR) technique. The measured values were corrected by simultaneously measured air pressure with a hermetically sealed nitrogen-filled gas cuvette due to no flowing reference gas used. Two commercially available working standards (310 and 380 ppm of $CO_2$ in $N_2$) were used for calibration every day at different hours. The $CO_2$ concentration in this gas bottle was compared in short intervals with a reference standard provided by UBA which was adjusted to the Keeling standard reference scale.

20   At ZUG the sampling line consisted of a stainless steel tube with an inner core of borosilicate glass and a cylindrical stainless steel top cup against intake of precipitation. The inlet with the structure of a small mast ended approximately 4 m on the top of the laboratory building, which is situated on the Zugspitze summit platform (see Fig. 1b). Inside the laboratory a turbine with a fast real-time fine control ensured a constant sample inflow of 500 l/min of in-situ air. The borosilicate glass tube (about 10 cm diameter) continued inside the laboratory, providing a number of outlets from where the instruments could
25   get the sample air for their own analyses. The measurement and calibration were performed with a URAS-3G device and an Ansyco mixing box. The mixing controller allowed automatic switching for up to four calibration gases and sampling air by a self-written calibration routine using Testpoint software. The linear two-point calibration enveloping the actual ambient values with low and high $CO_2$ concentrations was taken at every $25^{th}$ hour. Every six months the working standards were checked and re-adjusted, when required, to the standard reference scale by inter-comparison measurements with the station
30   standards…"

Do you use the data already selected (according to time of the day or other criteria) from previous site managers?

**Authors**: No, the data we used in this study were quality validated without application of any pre-selection procedures. Only obvious outliers, due to such as malfunction or power failure, were left out as mentioned in Section 2.2. Therefore we added in the manuscript,

"…The $CO_2$ data from these measurement sites and from Mount Zugspitze locations were considered as validated data set (Level 2: calibrated, screened, artefacts and outliers removed), without any further data processing prior to the selection of representative data…"

2.3. Offset adjustment: The offset between the two sites (ZPT and ZUG) is huge with a large dispersion (5 to 6 ppm). A more detailed analysis of this offset, looking at its variation in time (and instrumental change), as a function of the atmospheric pressure, or CO2 concentration must be provided.

I support the hypothesis that the carrier gas effect can explain most of the differences between ZPT and ZUG, since it is well known that the CO2 concentration in air when using N2 mixtures as references, is under-estimated by few ppm. However, a discussion on this issue must be provided by the authors, with references to previous studies based on similar NDIR instruments (e.g. Pearman et al., Tellus, 1975; Griffith et al., Tellus, 1982). Is the observed differences compatible with what we can expect considering the literature, and the atmospheric pressure at this altitude site?

**Authors**: Thank you very much for the helpful literature. We have made a more detailed description and analysis for the offset adjustment now in both the manuscript and supplement. Please check the following text.

In manuscript:

"…However, for the three-year parallel $CO_2$ measurements at ZPT and ZUG (1995–1997), clear offsets of $-5.8 \pm 0.4$ ppm ($CO_{2,\ ZPT}$ minus $CO_{2,\ ZUG}$, $1 \cdot SD$) were observed. The major reason for this bias is assumed to be the pressure-broadening effect in the used gas analyzers and the different gas mixtures used in the standards, $CO_2/N_2$ vs. $CO_2$/air, the so called "carrier gas correction (CGC)" (Bischof, 1975; Pearman and Garratt, 1975). It is known from previous studies that the measured $CO_2$ concentration, when using $CO_2/N_2$ mixtures as reference, is usually underestimated by several ppms for the URAS instruments, and such offsets vary from different types of analyzers (Pearman, 1977; Manning and Pohl; 1986). The carrier gas effect varies even between the same type of analyzer as well as with replacement of parts of the analyzer (Griffith et al., 1982; Kirk Thoning, personal communication, August 1, 2018). Due to lack of information and impossible on-site experiments with previous calibration standards, an offset adjustment to the $CO_2$ data set at ZPT was made for further analyses based on the offsets in data computed in the overlapping years instead of a physically derived correction. A single correction factor

$$G = 0.956 + 0.00017 \cdot C_{ZPT} \tag{1}$$

was applied to the ZPT data while $C_{ZPT}$ denotes the $CO_2$ concentrations at ZPT. Because of the same calibration mixtures, an additional adjustment was applied to the $CO_2$ concentrations at WNK by calculating the $CO_2$ differences between ZPT and

WNK. A detailed description on the offset adjustment of CGC with potential errors is given in the supplement. Two similar CGCs by Manning and Pohl (1986) at Baring Head, New Zealand and Cundari et al. (1990) at Mt. Cimone, Italy, were comparable in magnitude to our offset adjustment…"

In supplement:

5 **2. Offset adjustment**

**2.1. Offset adjustment background**

From the observed data for the three-year parallel $CO_2$ measurements at ZPT and ZUG (1995–1997) we obtain an offset of – 5.8 ± 0.4 ppm ($CO_{2, ZPT}$ minus $CO_{2, ZUG}$, $1 \cdot SD$). In the present situation, on-site corrections based on different calibration standards and different types of analysers are no longer possible. Therefore instead of a laboratory data based correction of
10 this offset, we performed an offset adjustment, which was based on the historical time series. Above all, depending on the existing information, we have to make the assumption that none of the following effects have been corrected beforehand at ZPT but at ZUG.

As mentioned in the paper, it is assumed that such a large offset (several ppm) is mostly influenced by the so-called "carrier gas effect" on the infrared gas analysis investigated by Bischof (1975) and Pearman and Garratt (1975). There a considerable
15 deviation was detected due to the pressure broadening effects on the different types of used gas analyser, and more importantly to the different carrier gases used in the standards, i.e. $CO_2/N_2$ mixtures vs. $CO_2$/air mixtures. In Table S2, it is shown that between ZPT and ZUG during 1995–1997, the same type of analysers (URAS 3G, Hartmann & Braun) were used, but however the calibration gases were different ($CO_2/N_2$ for ZPT and $CO_2$/natural air for ZUG). Experiments implied that the $CO_2$ concentration in air when using $CO_2/N_2$ mixtures as references is usually underestimated by several ppms for
20 the URAS instruments. On the other hand, the measurement of $CO_2$ concentration in air is not affected if $CO_2$/air mixtures were used as references. From Pearman (1977), we learnt that the potential carrier gas error could range from –4.9 to +3.8 ppm (8.7 ppm in absolute difference) depending on different analysers (Bischof, 1975; Pearman, 1977). Griffith (1982) showed that this can vary even between analysers of the same type.

**Table S1: Detailed description of atmospheric $CO_2$ measurement techniques (NDIR = Nondispersive infrared, GC = Gas
25 chromatography, and CRDS = Cavity ring-down spectroscopy).**

| ID | Time period | Instrument (Analytical method) | Scale | Calibration gas |
|----|-------------|-------------------------------|-------|-----------------|
| ZPT | 1981–1997 | 1981–1984: Hartmann & Braun URAS 2 (NDIR) | WMO X74 scale | $CO_2$ in $N_2$ |
| | | 1985–1988: Hartmann & Braun URAS 2T (NDIR) | | |
| | | 1989–1997: Hartmann & Braun URAS 3G (NDIR) | | |
| ZUG | 1995–2001 | Hartmann & Braun URAS 3G (NDIR) | WMO X85 scale | $CO_2$ in natural air |
| ZSF | 2001–2016 | 2001–2016: Hewlett Packard Modified HP 6890 Chem. station (GC) | WMO X2007 scale | $CO_2$ in natural air |
| | | 2012–2013: Picarro EnviroSense 3000i (CRDS) | | |
| WNK | 1981–1996 | Hartmann & Braun URAS 2T (NDIR) | WMO X74 scale | $CO_2$ in $N_2$ |

Pearman (1977) also mentioned that both the sign and magnitude of the carrier gas error depend on not only the configuration and model of analyser used, but also the ambient pressure at which measurements are made, i.e. the station altitude. With an altitude difference of around 1.6 km, a difference in carrier gas effect of ~0.6 ppm was found when measurements were made with a URAS 2 (Pearman and Garratt, 1975). At Mount Zugspitze, the altitude difference between ZPT and ZUG is approximately 250 m, and thus the carrier effect dependence on the ambient pressure is rather limited.

Another potential factor is the drying problem due to the varying water content as described in Reiter et al. (1986). By comparing an URAS 2T with a URAS 3G at another measurement station in Garmisch-Partenkirchen (GAP), the humidity-induced error ranged from the extreme conditions in summer (at most 6 ppm), to 2 ppm in winter. Pearman (1975) also addressed this problem as non-dispersive infrared gas analysers were influenced by water vapour in the air sample. The subsequent measurement must be corrected by multiplying the indicated concentration by $(1 + 1.61 * r)^{-1}$, where $r$ is the water vapour mass mixing ratio of the undried air. However, such error indicated that the measured $CO_2$ concentration would be overestimated when not corrected. Moreover, this error also decreases with altitude and will be less than the resolution of the NDIR analysers (approximately $\pm 0.2$ ppm) above about 8 km a.s.l. Regarding that the absolute water content for mountain stations is, on average, very low (for example at ZSF, the relative humidity in sampling air ranges between 2–10% in winter and approximately 27–32% in summer at 20°C), such an effect of drying the air sample prior to analysis was assumed to be minor for Mount Zugspitze.

**2.2. Offset adjustment at ZPT**

In order to make the offset adjustment, we follow the approach from Griffith (1982) and Griffith et al. (1982), together with comparing similar carrier gas correction cases done by Manning and Pohl (1986b) and Cundari et al. (1990). The general assumption is that the carrier gas correction (CGC) term is proportional to $CO_2$ concentration (Griffith, 1982; Manning and Pohl, 1986a). Carrier gas effects were determined experimentally by comparing analyser values (apparent $CO_2$ concentration $C_a$) with true (mano-metrically determined) $CO_2$ concentration (true $CO_2$ concentration $C_t$). Two terms were used here as the carrier gas shift ($\Delta$) and the correction factor ($G$).

$$\Delta = C_a - C_t \qquad (1)$$

$$G = C_t / C_a \qquad (2)$$

In our case, given that $CO_2$ measurements between ZUG and ZSF show a comparable result in 2001, and the altitude difference between ZSF and ZPT is only about 70 m a.s.l., we consider the $CO_2$ measurements at ZUG to be the true value ($C_{ZUG,t}$) and the $CO_2$ measurements at ZPT to be the apparent value ($C_{ZPT,a}$). Thus the offset can be expressed as (see Fig. S2a),

$$\Delta = C_{ZPT,a} - C_{ZUG,t} \qquad (3)$$

and hence the correction factor can be expressed as (see Fig. S2b),

$$G = C_{ZUG,t}/C_{ZPT,a}. \tag{4}$$

[Figure]

**Figure S2: a) Histogram for the offsets ($\Delta$) between CO$_2$ measurements at ZPT and ZUG for the period of 1995–1997. b) Histogram of the correction factor ($G$) between CO$_2$ measurements at ZPT and ZUG for the period of 1995–1997.**

We then plotted the computed correction factors $G$ with the apparent concentration at ZPT ($C_{ZPT,a}$) throughout the three years (1995–1997) in Fig. S3. A linear relationship can be observed but for a certain interval of the data a clear shift is noticed. Then we tried to divide the time blocks and took a closer look at when or how this shift takes place. We found out that this shift happened from November to December 1995, possibly due to instrumental setup changes. Figure S4 showed the time blocks before, during, and after. Nevertheless, by fitting linear regression nearly identical regression lines were produced for all three time blocks. At the CO$_2$ concentration of 360 ppm, the correction factors for the three time blocks were computed as 1.01728, 1.01684, and 1.0172 respectively, in terms of the adjusted values of 366.2208, 366.0624, 366.192 ppm with a span of ±0.08 ppm. Within the interval from 340 ppm to 370 ppm of atmospheric CO$_2$ concentrations, the same calculation applied shows an error range in the adjusted values from ±0.06 to ±0.09 ppm.

[Figure]

Figure S3: Computed correction factor $G$ against $CO_2$ concentrations at ZPT from 1995 to 1997.

[Figure]

Figure S4: Computed correction factor $G$ against $CO_2$ concentrations at ZPT from 1995 to 1997 with three separate time blocks.

Therefore, for the shifted time block (1995-11-01 to 1995-12-31), we used the correction factors by the linear regression function in Fig. S4b. Since the rest of the time blocks showed nearly identical results, we combined the data together and made a new linear regression. Based on this regression function, we made the following offset adjustment for all the remaining $CO_2$ data sets at ZPT (1981–1997) except for the two months in 1995, as shown below

$$G = 0.956 + 0.00017 \cdot C_{ZPT,a}. \tag{5}$$

And the adjusted $CO_2$ concentrations at ZPT can be expressed as

$$C_{ZPT,t} = C_{ZPT,a} \cdot G = C_{ZPT,a} \cdot \left(0.956 + 0.00017 \cdot C_{ZPT,a}\right). \tag{6}$$

[Figure]

**Figure S5: Computed correction factor $G$ against $CO_2$ concentrations at ZPT from two separate time blocks, used for offset adjustment on the $CO_2$ data set at ZPT.**

The reason we chose a single correction factor for most of the years is that, from the given comparison of the three separate time blocks, the error is small (less than 0.1 ppm). Therefore it is assumed that with different instruments used throughout the measurement periods the offsets remain small and hence relatively stable. Figure S5 also showed that the points were slightly off the regression line at both the head and tail even with $R^2 = 1$. This leads to errors of up to 0.2 ppm for a range of 338.32 to 385.69 ppm ($CO_2$ minimum and maximum at ZPT for this period), which agrees well with Griffith et al. (1982) as same errors of up to 0.2 ppm were detected for a range of 200 to 450 ppm. As a result, the offset adjustment of single correction factor is considered to be adequate.

In two similar cases, Manning and Pohl (1986b) showed the CGC at a concentration of 340 ppm for the URAS-2T analyser varied from 5.5 ppm to 3.2 ppm. With our correction factor function at the concentration of 340 ppm, the CGC turns out to be 4.7 ppm, which is in a good agreement. From another study by Cundari et al. (1990), by a least-square linear interpolation the experimentally determined means of the ratios were expressed by the following equation

$$\bar{G} = 1.0008 + 2.51 \cdot 10^{-5} \cdot C_a. \tag{7}$$

Given the described range of $C_a$ approximately from 320 to 360 ppm, the ratio varied from 1.008832 to 1.009836 which in terms of CGC the values changed 2.8 to 3.5 ppm. With the same described range, the CGC based on our regression function results in the values between 3.3 and 6.2 ppm.

**2.3. Offset adjustment at WNK**

Due to lack of information and no available comparable additional measurements at nearby locations, we decided to make a more general offset adjustment on $CO_2$ data at WNK based on the adjusted $CO_2$ data at ZPT because the same $CO_2/N_2$ mixtures were used for calibration (see Table S1). The time period of $CO_2$ measurements at WNK used in this study is 1981–1996, which is completely covered by $CO_2$ measurements at ZPT. We assume that the differences in $CO_2$ concentrations remain similarly before and after the offset adjustment, which means

$$C_{WNK,a} - C_{ZPT,a} \approx C_{WNK,t} - C_{ZPT,t}. \tag{8}$$

Therefore, the adjusted $CO_2$ concentrations at WNK can be expressed as

$$C_{WNK,t} = C_{ZPT,t} + \left( C_{WNK,a} - C_{ZPT,a} \right). \tag{9}$$

Finally the offset adjustment at WNK was done by calculating the differences in $CO_2$ concentrations between WNK and ZPT raw data and then adding it to the adjusted $CO_2$ concentrations at ZPT to compute the adjusted $CO_2$ concentrations at WNK.

**2.4 Offset adjustment error estimation (ZPT to ZUG)**

At the end, the maximum possible error should be estimated. Based on literature review, several additional factors which may contribute to it apart from carrier gas effect, pressure effect, and drying problem (varying water content) were listed as mentioned above.

- Absolute limit error on every single G ratio: 0.4 ppm (Cundari et al., 1990)
  - Station relative accuracy: $\pm$ 0.2 ppm (Pearman, 1975)
- Temperature effects: URAS analyzers are thermostated and small temperature variations, as are likely to occur, should not cause noticeable errors and thus can be neglected (Griffith et al., 1982).
- Leaking detectors: 0.4 ppm (+ 0.4 ppm) for URAS analyzers with different leaking scenarios (Griffith et al., 1982)
  - We assume that according to the applied quality standard from the former IFU (Fraunhofer Institute for Atmospheric Environmental Research, today KIT/IFU) the analyzers did not have a systematic leaking.
  - Further it is assumed, that the measurements did not have a drift in the data, because of continuous quality assurance for the former IFU.

Based on the given information about the measurements, we did a practically best possible description of obviously existing errors in the values. Please always keep in mind that this is an attempt and approach to make proper use of these historical data with given errors. Different time period, different types of analysers (also the same type), different used reference gases, or any potential replacement on the instruments and artefacts would introduce more errors to the offset adjustment. Caution should always be taken when using this combined data set. We would recommend contacting the data provider for more detailed discussion, whenever a detailed analysis requires reliable information.

I understand that you have not applied the offset correction (-0.11ppm) between ZUG and ZSF. Please make it clear.

**Authors**: Done. We added, "…Therefore, no adjustments regarding this offset were applied to the data sets…"

2.4. ADVS data selection: "The percentages of ADVS-selected data are … 13.5% for Zugspitze": have you merged all three Zugspitze stations together in this analysis? Does it mean there is none significant differences between them?

**Authors**: For the ADVS data selection, the three measurement locations at Mount Zugspitze were processed separately. Previously the results of selected percentage were computed after combining the three data sets together. Now we calculated the selected percentages separately as well as shown in the plot (see Fig. 2a). There are significant differences in the selected percentages at a 95% confidence interval among the three measurement locations. From the selected results, we can see different percentages of selected data at the three measurement locations, i.e. ZPT (9.9%) > ZSF (13.6%) > ZUG (19.5%). In that way we can detect the highest data variability at the pedestrian tunnel (ZPT) and the lowest variability at Zugspitze Summit (ZUG).

[Figure]

**Figure 2: a) Time series plot of 30-min averaged $CO_2$ concentrations measured at Mount Zugspitze (ZPT, ZUG, and ZSF) and Wank (WNK), and hourly averaged $CO_2$ concentrations measured at Schauinsland (SSL) and Mauna Loa (MLO) with ADVS-selected results. b) Detrended mean diurnal cycles with starting time windows (in grey) for ADVS data selection.**

Could you provide some statistics of the hours which are selected at Zugspitze as representative of the background according to ADVS method?

**Authors**: Yes, we added a graph of the resulting starting time window in Fig. 2b for each measurement site/location. A grey tone scale shows the frequency of ADVS-selected $CO_2$ data per hour in the total number of $CO_2$ data in Fig. 3. And a general discussion based on the diurnal variation is given in Sect 3.1, as the following.

[Figure]

**Figure 3: Frequency of the percentages of the number of ADVS-selected CO$_2$ data for each hour (0 to 23) in the total number of CO$_2$ data. In the shown greyscale grey means 1%, 2% and black means 3% of the data.**

"…The resulting ADVS-selected CO$_2$ data showed a clear linkage of the percentage of selected data and the altitude of the
5 measurement site. Among the continental stations, the percentage increased with altitude. Lower percentage indicates higher
data variability due to lower elevation and proximity to local sources and sinks. At Schauinsland, the percentage of CO$_2$ data
by the ADVS selection was 6.3% while the percentages at Mount Zugspitze reached 9.9% (ZPT), 19.5% (ZUG), and 13.6%
(ZSF), respectively. A moderate percentage of 6.3% was also derived at Mount Wank. However, regarding the elevated
mountain station Mauna Loa on the island of Hawaii, a much higher percentage (40.0%) of CO$_2$ data was selected by ADVS
10 as representative of its background concentration mainly due to the very limited nearby anthropogenic sources as well as
mostly clean, well-mixed air arriving there. A similar result for an island mountain station can be found in Yuan et al. (2018)
where a percentage of 36.2% was computed for the CO$_2$ measurements at the station Izaña on Tenerife Island (28°19′ N,
16°30′ E, 2373 m a.s.l.). This can also be explained by the detrended mean diurnal cycles shown in Fig. 2(b) and Fig. 3. The
mean diurnal cycle at MLO only exhibits a clear trough during daytime, especially starting from 12:00h local time (LT),
15 which is believed to be influenced by the vegetation activity (photosynthesis) in the surroundings. The same effect can be
seen at WNK and SSL, but with larger magnitudes and earlier occurrences of the minima because of their lower locations
closer to CO$_2$ sinks. In contrast, at these two sites the CO$_2$ maxima in the diurnal cycles were not as clearly noticeable as at
Mount Zugspitze due to anthropogenic sources and high biogenic respiration. At the three locations of Mount Zugspitze, the
CO$_2$ peaks in the mean diurnal cycles are driven by the late-morning convective upslope wind, which was relatively obvious
20 at both ZUG and ZSF. However, from the perspective of data selection, a significantly higher percentage of CO$_2$ data was
selected at ZSF compared with ZPT although there is only a small difference in altitude of around only 70 m. This proves
that ZSF is capable to capture more background conditions than ZPT during the day. Nevertheless, based on the starting time
window computed for ADVS selection, we found that, in general, most stations exhibited similar starting time windows
beginning around midnight and the ADVS data selection was applied systematically by including more data around these
25 hours (see Fig. 3), which confirmed our assumption of background conditions during midnight for the ADVS data selection
(Yuan et al., 2018)…"

2.5. STL decomposition Missing monthly values were substituted by spline interpolation: do you allow an interpolation of large data gaps like several consecutive months?

**Authors**: No, such large data gaps of several consecutive months are not allowed. And that is the reason why we previously decided to apply STL decomposition only on the original $CO_2$ data sets without ADVS data selection to evaluate the trend and seasonality. Besides, for the global trend data sets applied by the same STL decomposition technique, there are only monthly values available which cannot be selected by ADVS. Regarding the original $CO_2$ data sets, there is only one such large data gap for consecutive six months, which occurred at ZUG from July to December of 1998. Thus we performed the STL decomposition separately before and after this time period.

For improvement, we decided now to also apply STL decomposition to the ADVS-selected data at stations at Mount Zugspitze and Mauna Loa, as there are no large data gaps in the monthly averages from ADVS-selected data at these sites/locations. These results are implemented and discussed throughout the manuscript.

"…especially for measurement sites at lower elevations": I am confused about which sites you are referring to. Low altitude sampling locations at Zugspitze, or other sites like Schauinsland? Can you be more specific about the data gaps at stations, since it would make much more sense to use background data (after ADVS selection) for the seasonal and trend analysis, especially when comparing at other large scale time series.

**Authors**: More descriptions have been added in the text and also mentioned in the previous answer. Here the measurement sites at lower elevations refer to WNK and SSL. For a detailed illustration on each component of STL decomposition, we now included all the decomposed plots in Supplement S3.

3.1. Trend and seasonality "Only the mean annual growth rate between 1995 and 2001 at the ZUG site is much lower than the other sites due to missing values in 1998": Not clear for me why the 6 months data gap in summer 1998 decreases so much the total trend over the period 1995 to 2001. Please clarify.

**Authors**: Sorry for the confusion. What we want to point out is that because of the data gap in 1998 at ZUG, the annual growth rates of 1998 and 1999 are not accounted for in the mean annual growth rate calculation. However, for all of the other measurement sites, a clear anomalous peak in $CO_2$ annual growth rate is shown which can be attributed to a strong El Niño event. Therefore, we have rephrased this paragraph as the following.

"…This can be explained by the missing monthly values in 1998 and thus in turn the annual growth rates of 1998 and 1999 were left out for the average. However, the annual growth rates of these two years reached anomalous peaks at most sites (see details later in Sect. 3.6)…"

"Amplitudes of 15.44 and 14.89 ppm": For most signals I would suggest rounding the values to one decimal place.

**Authors**: Thank you very much for your suggestion. We have rounded all the values across the manuscript and rewrote the content accordingly.

The comparison of the seasonal cycles would be much more meaningful with background selected data. By the way do you use also all data (without selection) at SSL, WNK and MLO sites, or do you use the data selected by the station's managers at those sites? Please clarify. It could be interesting to see if the ZSF site remains more influenced by the air from the valleys, compared to ZPT and ZUG, once you have selected the nighttime values at all sites.

5   **Authors**: As mentioned above, we have now included the ADVS-selected data sets for the comparison of the seasonal cycles at Mount Zugspitze and Mauna Loa and similar results were found.

For sites SSL and WNK, all data were used for the analyses. There, no pre data selection routines have been performed. Therefore we added at the end of Sect. 2.2,

"…The $CO_2$ data from these measurement sites and from Mount Zugspitze locations were considered as validated data set
10   (Level 2: calibrated, screened, artefacts and outliers removed), without any further data processing prior to the selection of representative data…"

Regarding the comparison among ZSF, ZPT, and ZUG, the results of seasonal cycles are similar for the ADVS-selected data sets, that for ZSF clearly higher $CO_2$ levels were observed from January to March and lower $CO_2$ levels were observed from July to September (see Fig. 7).

[Figure]

**Figure 6: Mean $CO_2$ seasonal cycles from the STL seasonal component at each measurement site or location. Uncertainties at a 95% confidence interval are shown by the shaded areas with corresponding color.**

"there are slight differences in seasonal amplitudes (ZPT: 10.86 ppm; ZUG: 11.14 ppm; ZSF: 13.09 ppm) among the three sites": I would not call a 2 ppm signal a slight difference ! A major signal to look at for such long term time series in North
20   Hemisphere would be a possible trend in the amplitude of the seasonal cycle which could indicate a trend in the way the biosphere is interacting with atmospheric CO2. Graven et al., 2013 described for example increasing trends of the seasonal CO2 amplitude of 0.32 % per year at Mauna Loa and 0.60 % per year at Point Barrow. Considering a mean amplitude of about 12 ppm you could expect a trend of 1.4 to 2.5 ppm over the 36 years period of measurement at Zugspitze (assuming the MLO and BRW trends).

**Authors**: Thank you very much for the correction and information. We leave out the "slight" in the sentence. And with the new offset adjustment, the results are,

"…Despite the close proximity, there are differences in their seasonal amplitudes (ZPT: 11.9 ± 1.2 ppm; ZUG: 11.2 ± 1.0 ppm; ZSF: 13.3 ± 0.7 ppm). Good agreement is shown between $CO_2$ seasonal cycles from April to June and from October to December. However, significantly higher levels of $CO_2$ were evident at ZSF from January to March as well as lower levels from July to September. After data selection with lower seasonal amplitudes of 10.3 ± 1.3 ppm (ZPT_ADVS), 10.3 ± 1.2 ppm (ZUG_ADVS), and 10.9 ± 0.6 ppm (ZSF_ADVS), similar differences of the $CO_2$ levels in the seasonal cycles could be observed…"

Figure 3: the significant differences you show on figure 3b with the 3 sampling locations should prevent you from mixing those three dataset together as you do in figure 3a.

**Authors**: Thank you for the insight. As mentioned above, we have separated the three locations in all figures.

3.2. Inter-annual variations Abnormal high percentage at Zugspitze in 2000: I do not understand the sentence on line 5/6 suggesting that a careful and intensive selection was performed in 2000. Is the selection process different from the other years?

**Authors**: Sorry for the confusion. The original $CO_2$ data at ZUG was provided by the previous station manager Dr. H-E. Scheel, IFU. By direct cooperation, we learnt at that time that due to temporary systematic local effects of inflow of in-situ air to the sampling unit the $CO_2$ data at ZUG in 2000 had to be intensively selected by the operator. However the $CO_2$ data was only available in the format after this intensive selection so that such abnormal high percentage was derived.

Again, due to the differences between the three sampling locations (especially ZSF which is more influenced by air uplifted from the valleys) I think you should differentiate them in figure 4.

**Authors**: Done.

3.3. Weekly periodicity I would suggest to discuss short-term variabilities (weeks and daily) before trend and inter-annual variations.

**Authors**: Thank you for the suggestion. We have changed the order of the subsections in the results and discussion. Now we follow that,

- Sect. 3.1 ADVS selection and diurnal variation
- Sect. 3.2 Weekly periodicity
- Sect. 3.3 Case study on atmospheric CO, NO, and passenger numbers at Zugspitze
- Sect. 3.4 Trend
- Sect. 3.5 Seasonality

I do not see the interest of comparing the weekly variations at Zugspitze to the one observed at Mauna Loa.

**Authors**: We decided to keep the comparison of the weekly periodicity between Mount Zugspitze and Mauna Loa. The reason is that this method of calculating the MSR values for evaluating the weekly cycle was developed by using the Mauna Loa $CO_2$ data (Cerveny and Coakley, 2002). The results show different weekly characteristics between ZSF and MLO, but not for the previous time periods with ZPT and ZUG.

**References**

Bischof, W.: The influence of the carrier gas on the infrared gas analysis of atmospheric $CO_2$, Tellus, 27, 59–61, doi:10.3402/tellusa.v27i1.9884, 1975.

Cerveny, R. S. and Coakley, K. J.: A weekly cycle in atmospheric carbon dioxide, Geophys. Res. Lett., 29, 967, doi:10.1029/2001GL013952, 2002.

Cundari, V., Colombo, T., Papini, G., Benedicti, G., and Ciattaglia, L.: Recent improvements on atmospheric $CO_2$ measurements at Mt. Cimone observatory, Italy, Il Nuovo Cimento C, 13, 871–882, doi:10.1007/BF02512003, 1990.

Griffith, D. W. T.: Calculations of carrier gas effects in non-dispersive infrared analyzers I. Theory, Tellus, 34, 376–384, doi:10.1111/j.2153-3490.1982.tb01827.x, 1982.

Griffith, D. W. T., Keeling, C. D., Adams, A., Guenther, P. R., and Bacastow, R. B.: Calculations of carrier gas effects in non-dispersive infrared analyzers. II. Comparisons with experiment, Tellus, 34, 385–397, doi:10.3402/tellusa.v34i4.10825, 1982.

Manning, M. R. and Pohl, K. P.: A Review of $CO_2$ in Air Calibration Gas Mixtures used at Baring Head, New Zealand, Institute of Nuclear Sciences, DSIR, New Zealand, Report No INS-R--351, 1986a.

Manning, M. R. and Pohl, K. P.: Atmospheric $CO_2$ Monitoring in New Zealand 1971-1985, Institute of Nuclear Sciences, DSIR, New Zealand, Report No INS-R--350, 1986b.

Pearman, G. I.: A correction for the effect of drying of air samples and its significance to the interpretation of atmospheric $CO_2$ measurements, Tellus, 27, 311–317, doi:10.3402/tellusa.v27i3.9909, 1975.

Pearman, G. I.: Further studies of the comparability of baseline atmospheric carbon dioxide measurements, Tellus, 29, 171–181, doi:10.3402/tellusa.v29i2.11343, 1977.

Pearman, G. I. and Garratt, J. R.: Errors in atmospheric $CO_2$ concentration measurements arising from the use of reference gas mixtures different in composition to the sample air, Tellus, 27, 62–66, doi:10.3402/tellusa.v27i1.9885, 1975.

Reiter, R., Sladkovic, R., and Kanter, H.-J.: Concentration of trace gases in the lower troposphere, simultaneously recorded at neighboring mountain stations, Meteorl. Atmos. Phys., 35, 187–200, doi:10.1007/BF01041811, 1986.

---

## Author Comment (AC2) · 19 Dec 2018

*Author comments on* "**On the diurnal, weekly, seasonal cycles and annual trends in atmospheric $CO_2$ at Mount Zugspitze, Germany during 1981–2016**" *by* **Ye Yuan et al.**

Ye Yuan on behalf of all co-authors

Answers to **Anonymous Referee #2 (RC2)**

The referee comments are shown in black. The answers are shown in blue.

General Comments

10 This paper outlines a set of CO2 data records collected over >30 years at locations within the German alps, specifically the methods used and trends observed. These long-term continental records, although more complicated to interpret than coastal records, are important. As such details of these records, like those given in this paper, should be published and the records themselves made publicly available. Unfortunately, there is a distinct lack of detail when it comes to the calibration approach used, in particular for the older data records. This needs to be rectified before publication. The paper also has a number of

15 sentences which are confusing to read and would benefit greatly from the Copernicus copy editing service or the help of a native English speaker. I have attempted to note these in the technical corrections section and offered some suggestions for how they could be clarified. I feel that only with the addition of significant detail in relation to the calibration approach and a revision of the language used should the paper should be published.

**Authors:** We would like to thank Anonymous Referee #2 for the efforts to review this manuscript and to provide very

20 helpful comments and detailed remarks. All the referee's comments have been carefully examined and addressed in the revised manuscript as well as supplement. Besides, we have improved the manuscript with English proofreading.

Specific Comments

Abstract

Examining weekend-weekday variability in order to comment on which fluxes are driving CO2 signals is a powerful tool.

25 This, along with the outcomes of such a study should be highlighted in the abstract. At the moment the reference to it is rather vague, "indicating potential CO2 sources", and could easily be strengthened.

**Authors:** Thank you very much for the point. We have rephrased now in the abstract,

"…For a comprehensive site characterization of Mount Zugspitze, analyses of $CO_2$ weekly periodicity and diurnal cycle were performed to provide evidence for local sources and sinks, showing clear weekday to weekend differences with dominantly higher $CO_2$ levels during the daytime of the weekdays. A case study of atmospheric trace gases (CO and NO) and passenger numbers to the summit indicate that closeby $CO_2$ sources did not result from tourist activities but obviously from anthropogenic pollution in the near vicinity. Such analysis of local effects is an indispensable requirement for selecting representative data at orographic complex measurement sites…"

**2.1 Measurement sites**

There needs to be a description of the sampling method. Was there a small mast at these locations with an intake cup or were the instruments just measuring the air around them?

**Authors:** Done. We have included more details for the instrumental setup and restructured all the content of sampling method into section of instrumental setup and data processing.

Regarding sampling system, the text added in the manuscript is the following,

"…At ZUG the sampling line consisted of a stainless steel tube with an inner core of borosilicate glass and a cylindrical stainless steel top cup against intake of precipitation. The inlet with the structure of a small mast ended approximately 4 m on the top of the laboratory building, which is situated on the Zugspitze summit platform (see Fig. 1b). Inside the laboratory a turbine with a fast real-time fine control ensured a constant sample inflow of 500 l/min of in-situ air. The borosilicate glass tube (about 10 cm diameter) continued inside the laboratory, providing a number of outlets from where the instruments could get the sample air for their own analyses…"

"…At ZSF the same construction principle was applied for atmospheric sampling. There, the mast ends about 2.5 m above the pavement of the research terrace at the 5$^{th}$ floor in an altitude of 2670 m a.s.l…"

**2.2 Data processing**

If you're presenting data from the first two time periods then you need to give information on how that data (or wasn't calibrated). If they weren't calibrated then say so, and in the discussion provide an estimate of the size of the error that this will drive in the data. The GC calibration method is unclear to me. From the description it appears that you have a single working standard, the concentration of which is adjusted based on the station standards, and that this working standard is measured once every 15 mins. This will account for instrumental drift but does that mean you're assuming a linear detector response? Using GC to measure CO2 is usually a more linear approach than many other CO2 measurement techniques but it's not exactly linear. The effect of this non-linearity needs discussed and outlined in the text. There is no information on the CRDS calibration process. If CRDS data is presented in the paper (it's not clear if it is) then this information needs to be provided.

**Authors:** Thank you very much for your comment. We have now included a more detailed description on the instrumental setup for all measurement locations as mentioned in the previous answer. (This is the same as the comment regarding instrumental setup and data processing for Anonymous Referee #1.)

"…The $CO_2$ measurement at ZPT was continuously performed with different, consecutively used instrument models (i.e., the URAS-2, 2T, and 3G) of nondispersive infrared (NDIR) technique. The measured values were corrected by simultaneously measured air pressure with a hermetically sealed nitrogen-filled gas cuvette due to no flowing reference gas used. Two commercially available working standards (310 and 380 ppm of $CO_2$ in $N_2$) were used for calibration every day at different hours. The $CO_2$ concentration in this gas bottle was compared in short intervals with a reference standard provided by UBA which was adjusted to the Keeling standard reference scale…"

"…The measurement and calibration were performed with a URAS-3G device and an Ansyco mixing box. The mixing controller allowed automatic switching for up to four calibration gases and sampling air by a self-written calibration routine using Testpoint software. The linear two-point calibration enveloping the actual ambient values with low and high $CO_2$ concentrations was taken at every $25^{th}$ hour. Every six months the working standards were checked and re-adjusted, when required, to the standard reference scale by inter-comparison measurements with the station standards…"

Regarding the GC measurement, there has been used a working standard and a target. The working standard was supplied by a German specialist Deuste-Steiniger. Before practical use it has been measured approximately 800 to 1000 times against a group of 6 station reference standards, provided by NOAA for the time of 9 months. This longer time for intercomparison was needed to determine and exclude a possible drift of the standard and to adjust the $CO_2$ concentrations of the working standard as precisely as possible. The target was provided by the University of Heidelberg and had a slightly higher $CO_2$ concentration. The role of the target is to ensure a consistency of the measurement accuracy over the time. The target was measured every day about 25 times. The working standard was re-checked every two months with intercomparison measurements against the station reference standards from NOAA. If required, values of the measurements will have to be corrected. Actually the measurement of $CO_2$ is via a $CH_4$ equivalent by the use of FID. In the GC, the collected $CO_2$ is converted to $CH_4$ on a nickel catalyzer at a temperature of 400°C at the presence of hydrogen gas. The measurement of $CH_4$ with FID is known as linear and in this case no problems with non-linearity will occur.

2.3 Offset adjustment

The offset noted between ZPT and ZUG is very large – typically 6ppm – and concerning. However, it's difficult to comment on the offset adjustment used to correct for this as no information is given on how these sites are calibrated. Without further information it is impossible to know whether the offsets are driven solely by the use of CO2 in N2 calibration standards or other issues. It's also possible that, considering that they are different locations, that they were measuring air of different composition and part of this offset was true signal. Was any data filtering (e.g. wind speed/direction) completed prior to the comparison?

**Authors**: First of all, no pre-data filtering were done before the comparison. And we have now included a detailed analysis on the offset adjustment. Please see the following text. (This is the same as the comment regarding offset adjustment for Anonymous Referee #1.)

In manuscript:

5 "…However, for the three-year parallel $CO_2$ measurements at ZPT and ZUG (1995–1997), clear offsets of –5.8 ± 0.4 ppm ($CO_{2,\ ZPT}$ minus $CO_{2,\ ZUG}$, $1 \cdot SD$) were observed. The major reason for this bias is assumed to be the pressure-broadening effect in the used gas analyzers and the different gas mixtures used in the standards, $CO_2/N_2$ vs. $CO_2/air$, the so called "carrier gas correction (CGC)" (Bischof, 1975; Pearman and Garratt, 1975). It is known from previous studies that the measured $CO_2$ concentration, when using $CO_2/N_2$ mixtures as reference, is usually underestimated by several ppms for the

10 URAS instruments, and such offsets vary from different types of analyzers (Pearman, 1977; Manning and Pohl; 1986). The carrier gas effect varies even between the same type of analyzer as well as with replacement of parts of the analyzer (Griffith et al., 1982; Kirk Thoning, personal communication, August 1, 2018). Due to lack of information and impossible on-site experiments with previous calibration standards, an offset adjustment to the $CO_2$ data set at ZPT was made for further analyses based on the offsets in data computed in the overlapping years instead of a physically derived correction. A single

15 correction factor

$$G = 0.956 + 0.00017 \cdot C_{ZPT} \qquad\qquad (1)$$

was applied to the ZPT data while $C_{ZPT}$ denotes the $CO_2$ concentrations at ZPT. Because of the same calibration mixtures, an additional adjustment was applied to the $CO_2$ concentrations at WNK by calculating the $CO_2$ differences between ZPT and WNK. A detailed description on the offset adjustment of CGC with potential errors is given in the supplement. Two similar CGCs by Manning and Pohl (1986) at Baring Head, New Zealand and Cundari et al. (1990) at Mt. Cimone, Italy, were

20 comparable in magnitude to our offset adjustment…"

In supplement:

**2. Offset adjustment**

**2.1. Offset adjustment background**

From the observed data for the three-year parallel $CO_2$ measurements at ZPT and ZUG (1995–1997) we obtain an offset of –

25 5.8 ± 0.4 ppm ($CO_{2,\ ZPT}$ minus $CO_{2,\ ZUG}$, $1 \cdot SD$). In the present situation, on-site corrections based on different calibration standards and different types of analysers are no longer possible. Therefore instead of a laboratory data based correction of this offset, we performed an offset adjustment, which was based on the historical time series. Above all, depending on the existing information, we have to make the assumption that none of the following effects have been corrected beforehand at ZPT but at ZUG.

As mentioned in the paper, it is assumed that such a large offset (several ppm) is mostly influenced by the so-called "carrier gas effect" on the infrared gas analysis investigated by Bischof (1975) and Pearman and Garratt (1975). There a considerable deviation was detected due to the pressure broadening effects on the different types of used gas analyser, and more importantly to the different carrier gases used in the standards, i.e. $CO_2/N_2$ mixtures vs. $CO_2$/air mixtures. In Table S2, it is shown that between ZPT and ZUG during 1995–1997, the same type of analysers (URAS 3G, Hartmann & Braun) were used, but however the calibration gases were different ($CO_2/N_2$ for ZPT and $CO_2$/natural air for ZUG). Experiments implied that the $CO_2$ concentration in air when using $CO_2/N_2$ mixtures as references is usually underestimated by several ppms for the URAS instruments. On the other hand, the measurement of $CO_2$ concentration in air is not affected if $CO_2$/air mixtures were used as references. From Pearman (1977), we learnt that the potential carrier gas error could range from –4.9 to +3.8 ppm (8.7 ppm in absolute difference) depending on different analysers (Bischof, 1975; Pearman, 1977). Griffith (1982) showed that this can vary even between analysers of the same type.

**Table S1: Detailed description of atmospheric $CO_2$ measurement techniques (NDIR = Nondispersive infrared, GC = Gas chromatography, and CRDS = Cavity ring-down spectroscopy).**

| ID | Time period | Instrument (Analytical method) | Scale | Calibration gas |
|----|-------------|-------------------------------|-------|-----------------|
| ZPT | 1981–1997 | 1981–1984: Hartmann & Braun URAS 2 (NDIR)
 1985–1988: Hartmann & Braun URAS 2T (NDIR)
 1989–1997: Hartmann & Braun URAS 3G (NDIR) | WMO X74 scale | $CO_2$ in $N_2$ |
| ZUG | 1995–2001 | Hartmann & Braun URAS 3G (NDIR) | WMO X85 scale | $CO_2$ in natural air |
| ZSF | 2001–2016 | 2001–2016: Hewlett Packard Modified HP 6890 Chem. station (GC)
 2012–2013: Picarro EnviroSense 3000i (CRDS) | WMO X2007 scale | $CO_2$ in natural air |
| WNK | 1981–1996 | Hartmann & Braun URAS 2T (NDIR) | WMO X74 scale | $CO_2$ in $N_2$ |

Pearman (1977) also mentioned that both the sign and magnitude of the carrier gas error depend on not only the configuration and model of analyser used, but also the ambient pressure at which measurements are made, i.e. the station altitude. With an altitude difference of around 1.6 km, a difference in carrier gas effect of ~0.6 ppm was found when measurements were made with a URAS 2 (Pearman and Garratt, 1975). At Mount Zugspitze, the altitude difference between ZPT and ZUG is approximately 250 m, and thus the carrier effect dependence on the ambient pressure is rather limited.

Another potential factor is the drying problem due to the varying water content as described in Reiter et al. (1986). By comparing an URAS 2T with a URAS 3G at another measurement station in Garmisch-Partenkirchen (GAP), the humidity-induced error ranged from the extreme conditions in summer (at most 6 ppm), to 2 ppm in winter. Pearman (1975) also addressed this problem as non-dispersive infrared gas analysers were influenced by water vapour in the air sample. The subsequent measurement must be corrected by multiplying the indicated concentration by $(1 + 1.61 * r)^{-1}$, where $r$ is the water vapour mass mixing ratio of the undried air. However, such error indicated that the measured $CO_2$ concentration would be overestimated when not corrected. Moreover, this error also decreases with altitude and will be less than the resolution of the NDIR analysers (approximately ±0.2 ppm) above about 8 km a.s.l. Regarding that the absolute water content for mountain stations is, on average, very low (for example at ZSF, the relative humidity in sampling air ranges between 2–10%

in winter and approximately 27–32% in summer at 20°C), such an effect of drying the air sample prior to analysis was assumed to be minor for Mount Zugspitze.

**2.2. Offset adjustment at ZPT**

In order to make the offset adjustment, we follow the approach from Griffith (1982) and Griffith et al. (1982), together with comparing similar carrier gas correction cases done by Manning and Pohl (1986b) and Cundari et al. (1990). The general assumption is that the carrier gas correction (CGC) term is proportional to $CO_2$ concentration (Griffith, 1982; Manning and Pohl, 1986a). Carrier gas effects were determined experimentally by comparing analyser values (apparent $CO_2$ concentration $C_a$) with true (mano-metrically determined) $CO_2$ concentration (true $CO_2$ concentration $C_t$). Two terms were used here as the carrier gas shift ($\Delta$) and the correction factor ($G$).

$$\Delta = C_a - C_t \tag{1}$$

$$G = C_t / C_a \tag{2}$$

In our case, given that $CO_2$ measurements between ZUG and ZSF show a comparable result in 2001, and the altitude difference between ZSF and ZPT is only about 70 m a.s.l., we consider the $CO_2$ measurements at ZUG to be the true value ($C_{ZUG,t}$) and the $CO_2$ measurements at ZPT to be the apparent value ($C_{ZPT,a}$). Thus the offset can be expressed as (see Fig. S2a),

$$\Delta = C_{ZPT,a} - C_{ZUG,t} \tag{3}$$

and hence the correction factor can be expressed as (see Fig. S2b),

$$G = C_{ZUG,t} / C_{ZPT,a}. \tag{4}$$

[Figure]

**Figure S2:** a) Histogram for the offsets ($\Delta$) between $CO_2$ measurements at ZPT and ZUG for the period of 1995–1997. b) Histogram of the correction factor ($G$) between $CO_2$ measurements at ZPT and ZUG for the period of 1995–1997.

We then plotted the computed correction factors $G$ with the apparent concentration at ZPT ($C_{ZPT,a}$) throughout the three years (1995–1997) in Fig. S3. A linear relationship can be observed but for a certain interval of the data a clear shift is noticed. Then we tried to divide the time blocks and took a closer look at when or how this shift takes place. We found out that this shift happened from November to December 1995, possibly due to instrumental setup changes. Figure S4 showed the time blocks before, during, and after. Nevertheless, by fitting linear regression nearly identical regression lines were produced for all three time blocks. At the $CO_2$ concentration of 360 ppm, the correction factors for the three time blocks were computed as 1.01728, 1.01684, and 1.0172 respectively, in terms of the adjusted values of 366.2208, 366.0624, 366.192 ppm with a span of ±0.08 ppm. Within the interval from 340 ppm to 370 ppm of atmospheric $CO_2$ concentrations, the same calculation applied shows an error range in the adjusted values from ±0.06 to ±0.09 ppm.

[Figure]

Figure S3: Computed correction factor $G$ against $CO_2$ concentrations at ZPT from 1995 to 1997.

[Figure]

Therefore, for the shifted time block (1995-11-01 to 1995-12-31), we used the correction factors by the linear regression function in Fig. S4b. Since the rest of the time blocks showed nearly identical results, we combined the data together and made a new linear regression. Based on this regression function, we made the following offset adjustment for all the remaining $CO_2$ data sets at ZPT (1981–1997) except for the two months in 1995, as shown below

$$G = 0.956 + 0.00017 \cdot C_{ZPT,a}. \tag{5}$$

And the adjusted $CO_2$ concentrations at ZPT can be expressed as

$$C_{ZPT,t} = C_{ZPT,a} \cdot G = C_{ZPT,a} \cdot \left(0.956 + 0.00017 \cdot C_{ZPT,a}\right). \tag{6}$$

**1995-01-01 to 1995-10-31 and 1996-01-01 to 1997-12-31**

**Figure S5: Computed correction factor $G$ against $CO_2$ concentrations at ZPT from two separate time blocks, used for offset adjustment on the $CO_2$ data set at ZPT.**

The reason we chose a single correction factor for most of the years is that, from the given comparison of the three separate time blocks, the error is small (less than 0.1 ppm). Therefore it is assumed that with different instruments used throughout the measurement periods the offsets remain small and hence relatively stable. Figure S5 also showed that the points were slightly off the regression line at both the head and tail even with $R^2 = 1$. This leads to errors of up to 0.2 ppm for a range of 338.32 to 385.69 ppm ($CO_2$ minimum and maximum at ZPT for this period), which agrees well with Griffith et al. (1982) as same errors of up to 0.2 ppm were detected for a range of 200 to 450 ppm. As a result, the offset adjustment of single correction factor is considered to be adequate.

In two similar cases, Manning and Pohl (1986b) showed the CGC at a concentration of 340 ppm for the URAS-2T analyser varied from 5.5 ppm to 3.2 ppm. With our correction factor function at the concentration of 340 ppm, the CGC turns out to

be 4.7 ppm, which is in a good agreement. From another study by Cundari et al. (1990), by a least-square linear interpolation the experimentally determined means of the ratios were expressed by the following equation

$$\bar{G} = 1.0008 + 2.51 \cdot 10^{-5} \cdot C_a. \tag{7}$$

Given the described range of $C_a$ approximately from 320 to 360 ppm, the ratio varied from 1.008832 to 1.009836 which in terms of CGC the values changed 2.8 to 3.5 ppm. With the same described range, the CGC based on our regression function results in the values between 3.3 and 6.2 ppm.

**2.3. Offset adjustment at WNK**

Due to lack of information and no available comparable additional measurements at nearby locations, we decided to make a more general offset adjustment on $CO_2$ data at WNK based on the adjusted $CO_2$ data at ZPT because the same $CO_2/N_2$ mixtures were used for calibration (see Table S1). The time period of $CO_2$ measurements at WNK used in this study is 1981–1996, which is completely covered by $CO_2$ measurements at ZPT. We assume that the differences in $CO_2$ concentrations remain similarly before and after the offset adjustment, which means

$$C_{WNK,a} - C_{ZPT,a} \approx C_{WNK,t} - C_{ZPT,t}. \tag{8}$$

Therefore, the adjusted $CO_2$ concentrations at WNK can be expressed as

$$C_{WNK,t} = C_{ZPT,t} + \left(C_{WNK,a} - C_{ZPT,a}\right). \tag{9}$$

Finally the offset adjustment at WNK was done by calculating the differences in $CO_2$ concentrations between WNK and ZPT raw data and then adding it to the adjusted $CO_2$ concentrations at ZPT to compute the adjusted $CO_2$ concentrations at WNK.

**2.4 Offset adjustment error estimation (ZPT to ZUG)**

At the end, the maximum possible error should be estimated. Based on literature review, several additional factors which may contribute to it apart from carrier gas effect, pressure effect, and drying problem (varying water content) were listed as mentioned above.

- Absolute limit error on every single G ratio: 0.4 ppm (Cundari et al., 1990)
  - Station relative accuracy: ± 0.2 ppm (Pearman, 1975)
- Temperature effects: URAS analyzers are thermostated and small temperature variations, as are likely to occur, should not cause noticeable errors and thus can be neglected (Griffith et al., 1982).
- Leaking detectors: 0.4 ppm (+ 0.4 ppm) for URAS analyzers with different leaking scenarios (Griffith et al., 1982)
  - We assume that according to the applied quality standard from the former IFU (Fraunhofer Institute for Atmospheric Environmental Research, today KIT/IFU) the analyzers did not have a systematic leaking.

o    Further it is assumed, that the measurements did not have a drift in the data, because of continuous quality assurance for the former IFU.

Based on the given information about the measurements, we did a practically best possible description of obviously existing errors in the values. Please always keep in mind that this is an attempt and approach to make proper use of these historical data with given errors. Different time period, different types of analysers (also the same type), different used reference gases, or any potential replacement on the instruments and artefacts would introduce more errors to the offset adjustment. Caution should always be taken when using this combined data set. We would recommend contacting the data provider for more detailed discussion, whenever a detailed analysis requires reliable information.

Technical and editorial corrections

The below comments are made in reference to specific areas of the text identified as page no./line no.

1/17 In this context there is no need for the definite article before "Mauna Loa" and "global means". This error occurs throughout text. For example "in good agreement with the Mauna Loa station and the global means" should read "in good agreement with Mauna Loa and global means"

**Authors:** Thank you very much for pointing this out. We have changed it throughout the manuscript.

1/18 It's important to include some estimate of the variability of the seasonal amplitude to give an indication of how stable it is.

**Authors:** Thank you very much for the suggestion. Now we have included the variability for the seasonal amplitude throughout the manuscript.

"…The peak-to-trough difference of the mean $CO_2$ seasonal cycle is 12.4 ± 0.6 ppm at Mount Zugspitze (after data selection: 10.5 ± 0.5 ppm), which is much lower than at nearby measurement sites at Mount Wank (15.9 ± 1.5 ppm) and Schauinsland (15.9 ± 1.0 ppm), but following a similar seasonal pattern…"

1/20-22 This sentence is confusing and vague.

**Authors:** We have rephrased it as following,

"…For a comprehensive site characterization of Mount Zugspitze, analyses of $CO_2$ weekly periodicity and diurnal cycle were performed to provide evidence for local sources and sinks, showing clear weekday to weekend differences with dominantly higher $CO_2$ levels during the daytime of the weekdays. A case study of atmospheric trace gases (CO and NO) and passenger numbers to the summit indicate that closeby $CO_2$ sources did not result from tourist activities but obviously from anthropogenic pollution in the near vicinity. Such analysis of local effects is an indispensable requirement for selecting representative data at orographic complex measurement sites…"

1/31-2/1 Please change "Apart from the sites located either in the Antarctica or along coastal/island regions, continental mountain stations also offer excellent options to observe the background atmospheric levels due to high elevations that are least unaffected…" to "Along with sites located in Antarctica or along coastal/island regions, continental mountain stations offer excellent options to observe background atmospheric levels due to high elevations that are less affected…"

5 **Authors:** Done.

2/2-4 This sentence is superfluous, please remove. "Presently, there are 31 Global Observatories coordinated by the Global Atmosphere Watch (GAW) network, focusing on monitoring the physical and chemical state of the atmosphere on a global scale."

**Authors:** Done.

10 2/7 Please change "lidar" to "LIDAR" it's an acronym.

**Authors:** Done.

2/15 Change "…what extend that elevated…" to "…what extent elevated…"

**Authors:** Done.

2/33 Confusing "Weekly CO2 periodicities were evaluated with the diurnal cycles for the Mount Zugspitze sites". Do you
15 mean that the weekly periodicity was evaluated by examining diurnal cycles or that the weekly periodicity was evaluated and diurnal cycles were also evaluated? I think the former but it could be read both ways.

**Authors:** We have rephrased the sentence as,

"…Short-term variations of weekly $CO_2$ periodicities and diurnal cycles were evaluated for Mount Zugspitze…"

3/1-2 Again "In addition, we perform an atmospheric CO and NO case study together with the amount of passengers at
20 Zugspitze in 2016 as potential indicators for weekday–weekend influences." is confusing. I'm guessing you mean "A case study combining atmospheric CO and NO measurements and records of passenger numbers was used to examine weekday-weekend differences"?

**Authors:** Thank you very much. It has been rephrased.

3/8-11 This is confusing. Please change "The measurements were collected at a southward-facing balcony in a pedestrian
25 tunnel (Reiter et al., 1986) from the summit of Mount Zugspitze to the Schneefernerhaus (ZPT, 4725  š N, 1059  š E, slightly below the summit), which was a hotel until 1992 when it was rebuilt into an environmental research station. From 1995 until 2001, a new set of measurements began at the summit (ZUG, 4725  š N, 1059  š E, 2960 m a.s.l.) at a sheltered laboratory on the terrace using a URAS-3G device." to "The measurements were collected at a southward-facing balcony of a pedestrian tunnel (Reiter et al., 1986) which joined the summit of Mount Zugspitze to the Schneefernerhaus

situated Xm below the summit (ZPT, 4725âAˇ š N, 1059âAˇ š E). The Schneefernerhaus was a hotel until 1992 when it was rebuilt into an environmental research station. From 1995 until 2001, a new set of measurements were made at a sheltered laboratory on the terrace of the summit (ZUG, 4725âAˇ š N, 1059âAˇ š E, 2960 m a.s.l.) using a URAS-3G device."

**Authors:** Done.

5   3/15-18 This section ("Zugspitzplatt, a glacier … shown in Fig 1. (Gantner et al., 2003)" interrupts the flow of the site descriptions. It's also unclear why it's included – I'm guessing to highlight that there are visitors nearby? Please move it to the end of the paragraph and provide more context.

**Authors:** We have re-structured this section. This section now only describes about the surrounding environment of the measurement locations. More detailed descriptions about instrumental setup and data processing were moved to the next
10  section.

3/20-21 Confusing. Were the CRDS measurements made as well as the GC measurements i.e in parallel? Or instead of due to the instrumental failure? It's unclear.

**Authors:** We have rephrased this information in the data processing. The CRDS measurements started in 2011 and were performed in parallel with the GC system. We only use the CRDS data for 2012 and 2013 because GC data were not
15  available.

"…Measurements of $CO_2$ at Schneefernerhaus continued thereafter to the present with a modified HP 6890 by using gas chromatography (GC) with an intermediate upgrade in 2008 (Bader, 2001; Hammer et al., 2008; Müller, 2009). In 2012 and 2013, because of an instrumental failure of the GC, $CO_2$ data were recorded with a cavity ring-down spectrometer (CRDS, Picarro EnviroSense 3000i) connected to the same air inlet, which had been installed in parallel since 2011…"

20  4/3 What were the concentrations of the working standards? I don't need the exact value for each cylinder but a general description would be useful. E.g. "near-ambient"

**Authors:** Thank you for the remark. "Near-ambient" was added.

4/5-6 Confusing. The GC data acquisition system doesn't "produce" the calibration values. By their very definition acquisition systems can only acquire data. Do you mean that using the GC system chromatograms were measured every 5
25  minutes with the working standard measured every third chromatogram?

**Authors:** The HP6890 GC measurement takes five minutes for one chromatogram. The typical sequence is sample, sample standard. With the chemstation software an automated realtime integration of chromatogram peaks was performed continuously. Together with the GC organizer software of the University of Heidelberg every two to four days the calculation of in situ $CO_2$ concentrations was performed. For continuous quality assurance the GC was checked daily for
30  flows, retention times, gas pressures, and the structure of chromatograms.

4/8 What is a "pollution list"?

**Authors:** In the Environmental Research Station Schneefernerhaus we have a "central logbook for local pollution from working activities in the Research Station". It is a strict rule, that every worker, crafter or colleague writes in the start- and end-time and the activity. This enables the scientists to do a well-organized data flagging of time sequences with air pollution.

4/8 "Simultaneous measurements of identical gas" Do you really mean that you have simultaneous measurements of CO2 made using another instrument at the same location? If so how were they made and why aren't they reported here?

**Authors:** As mentioned above, a second CRDS measurement started in 2011 in parallel. But we only used the CRDS data of $CO_2$ for 2012 and 2013 due to the instrumental failure of GC.

4/12 If the working standard is measured every 15 minutes how often was the second target measured?

**Authors:** Every day for about 25 times.

4/18 This is a really large offset, typically 6ppm. Please give the mean offset here so that readers don't have to look in the supplementary.

**Authors:** Done.

5/1-2 It would be useful to refer to this 36-year data record as a "compound" data record as it's actually composed of data collected at three different locations. Using this terminology would make later sections of paper clearer.

**Authors:** Done. We have rephrased this combing the comment from Anonymous Referee #1, using "composite".

"…In this study, we took $CO_2$ measurements during the corresponding time intervals at ZPT (1981–1994), ZUG (1995–2001), and ZSF (2002–2016) to assemble a composite time series for Mount Zugspitze over 36 years…"

6/11 Was this done on the raw data or the ADVS filtered data?

**Authors:** The MSR weekly periodicity analysis was done on the calibrated and quality assured data set (Level 2), which here is named "raw data" that have not been selected by ADVS.

6/23 Change "over the entire 36 year period" to "of the 36-year compound record"

**Authors:** Done.

6/25-26 "In general, the mean annual growth rates over the entire 36 year period at all sites agree within a range of 1.8 ppm yr–1". Which sites are you referring to here? The Zugspitze sites don't cover a 36-year period e.g. ZPT is only 16 years long. If you're referring to SSl, MLO and the global mean as referenced in the previous sentence than this sentence is redundant please remove it.

**Authors:** This refers to the 36-yr composite record of atmospheric $CO_2$ at Mount Zugspitze. The reason is that we have done the offset adjustment between ZPT and ZUG, and also the offset between ZUG and ZSF in 2001 is within ±0.1 ppm. This evidence makes us think that it is applicable to compare the mean annual growth rate of overall 36 years with other measurement stations. Of course, we also showed the mean annual growth rate at each measurement locations at each separate time blocks.

7/1-2 Please change "Möller (2017) also mentioned that growth rates at both German stations and the MLO from 1981 to 1992 were identical." To "Möller (2017) also mentioned that 1981 to 1992 growth rates at both German stations and MLO were identical."

**Authors:** Done.

7/8 Please change "that minimize in August" to "that reach a minimum in August".

**Authors:** Done.

7/10-11 Please change "Sampled air is more frequently mixed with air from lower levels, which is characterized by lower CO2 concentrations that also minimize in August." To "As such, in Summer sampled air is more frequently mixed with air from lower levels, which is characterized by lower CO2 concentrations, enhancing the August minimum."

**Authors:** Done.

7/17 Please change "The MLO is" to "Mauna Loa data are" or "The Mauna Loa CO2 record is"

**Authors:** Done.

7/18 Please change "which agree" to "which agrees"

**Authors:** Done.

7/18-19 Please change "Moreover, global means exhibited the lowest seasonal amplitudes of 4.33 ppm (NOAA) and 4.76 ppm (WDCGG)." To "Global means exhibited the lowest seasonal amplitudes, 4.33 ppm (NOAA) and 4.76 ppm (WDCGG)."

**Authors:** Done.

7/19-23 I know what you're trying to say but this section really isn't written clearly. Please correct it.

**Authors:** Done. We have rephrased it as,

"…Compared with WDCGG, NOAA global mean fits better the seasonal cycle of MLO  supporting the presence of a typical Marine Boundary Layer (MBL) condition for the levels of background $CO_2$ in the atmosphere. On the other hand, the WDCGG global mean includes continental characteristics for its calculation, thus exhibiting a slightly more continental signature which can be equally seen in the seasonal cycles at continental sites, such as Mount Zugspitze…"

7/27-28 Please change "Apart from this, significantly higher levels of CO2 at ZSF from January to March and lower levels from July to September cannot be neglected." To "However, significantly higher levels of CO2 are evident at ZSF from January to March and lower levels from July to September."

**Authors:** Done.

8/6-7 I'm confused. You state that there are an abnormally high percentage of validated data points for the year 2000 but then say there are only 4634 points but there are 15000 for the other years. Do you mean 15000 total for the remaining years or 15000 per year? If it's per year then that's seems wrong.

**Authors:** Sorry for the misunderstanding. The abnormally high percentage refers to the percentage of ADVS-selected data in the validated data in that year. At ZUG in 2000, an intensive data filtering had to be performed. Hence, the number of validated data points is much lower than in other years. But because this intensive data filtering resulted in a comparably low data variability, the data selection ADVS gave a considerably better relative percentage of representative data in 2000.

8/11-12 In figure 4b please colour code the sections of the compound Mt Zugspitze record for the different sites to make it easy to identify which years are ZPT, ZUG or ZSF. This would make relating this section to the figure far easier.

**Authors:** Done. Now we have separated these three measurement locations in all figures.

8/20 Please change "can also be illustrated for" to "are also evident in"

**Authors:** Done.

11/1-2 "Seasonal amplitude at … compared with global sites" This sentence doesn't make sense. Please correct.

**Authors:** Done. We have rephrased as,

"…Regarding the seasonal amplitude, Mount Zugspitze is significantly more influenced by biogenic activity, mostly in the summer compared with Mauna Loa and global means…"

Figure 2 Please plot the data from the different sites as different colours in the bottom left hand plot to make it clear which site is being used at which time.

**Authors:** Done. We have separated the three measurement locations.

Figure 4 – Please add the abbreviations used in the text e.g. SSL or WNK to the titles of the plots to make comparisons between the text and the figure easier. Please colour code the sections of the compound Mt Zugspitze record for the different sites to make it easy to identify which years are ZPT, ZUG or ZSF.

**Authors:** Done. We have changed the labels in the figures to the abbreviations used in the text and separate the Zugspitze measurement locations.

Figure 5 – Match the site colour coding from figure 4 to this figure.

**Authors:** Done.

---

## Author Comment (AC3) · 19 Dec 2018

Dear Mr. Ray Nassar,

thank you very much for your insightful suggestions regarding the additional literature. We have included them in the introduction now.

Best regards,

Ye Yuan
* * *
[Figure]

2018.

---

## Editor Decision (ED1)

Dear Ye Yuan and co-authors,

Thank-you for the work that you have undertaken to address the reviewers comments on your manuscript. Overall I think the manuscript is now close to being suitable for publication. There are some occasions where you have given a reply to the reviewer comment but do not appear to have captured that information sufficiently in the revised manuscript. I have listed these below. In reading the manuscript again, I would also like to recommend the following technical corrections. The page and line numbers refer to the version of the manuscript which includes the tracked changes.

P3, line 8: replace 'combing' with 'combining'

P4, line 25: perhaps replace 'against' with 'to prevent'

P4, line 25: Suggest replace the sentence starting 'The inlet …' with 'The inlet was mounted on a small mast (approximately 4 m high) on the top of the laboratory building …'

P5, line 3: Replace 'the mast ends' with 'the mast height is'

P5, line 13-14: Perhaps include the sentence from your reply to reviewer 2 (4/5-6): 'For continuous quality assurance the GC was checked daily for flows, retention times, gas pressures, and the structure of chromatograms.' somewhere in this paragraph.

P5, line 13: Replace 'pollution list' with 'logged list of local pollution from working activities in the Research Station.'

P5, line 18-19: Suggest replacing 'a quasi-continuously measured second target' with 'a second target (measured approximately 25 times per day)'

P6, line 2-3: You mention the different calibration scales here, but the only information about which scale is used for which time period is in the supplementary material (Table S1). I think it might be useful to move this table back into the manuscript, perhaps with a sentence at the end of section 2.2 such as 'The different instruments and calibration scales used at each location are summarised in Table 1.' For ZSF, perhaps you should be more explicit about the HP not being available in 2012-2013, and also that the CDRS has run for longer than 2012-2013, but those are the only years used in this study. This information could be added as table footnotes.

P6, line 7: Add '(Table 1)' after 'in the standards', assuming the instrument table is moved back to the main manuscript, otherwise refer to Table S1.

P6, line 13-15: Suggest replace sentence starting 'Due to ..' with 'Since we have insufficient information to determine a physically derived correction to the ZPT $CO_2$ data, an offset adjustment was made for further analyses based on the offsets in data computed in the overlapping years.'

P7, line 17: Delete 'results' before 'represent'.

P7, line 30: Suggest adding '(without ADVS selection)' after 'are calculated'

P11, line 21: Check table numbering if move table 1 back from supplement.

P12, line 19 and 20: To be consistent, you might want to change 'amplitude' to 'difference' in these sentences.

P12, last paragraph: Reviewer 1 suggests that one factor in differences in seasonal amplitude between Mount Zugspitze locations is the different measurement time periods and the possibility

that seasonal amplitude has grown over time. This is probably worth noting in this paragraph, alongside the discussion of different air-mass transport to the different locations.

P13, line 13: Suggest adding 'original' before 'validated' and 'only' before '4634'.

P16, line 2: Add 'composite' before 'CO2 measurement record'

Figure 2 caption: For panel (a) the caption needs to state that grey and black are used for the unselected and selected results.

Figure 3: Please consider whether this figure would be easier to understand if the percentages were calculated relative to the number of data points available for that hour, rather than to the total number of data. This would presumably give percentages that would average (rather than sum) to the number quoted in Sec 3.1.

Supplement, p2, line 16: Should be Table S1 not Table S2 (or Table 1) if this is moved into the paper as suggested above.

Supplement, p7, line 4: If move table, replace 'Table S1' with 'Table 1'

Supplement, Figure S7: You seem to have some odd behaviour at the start of the ZPT_ADVS timeseries with very low $CO_2$ concentrations. How has this occurred and will it have impacted any of your analysis?

Regards,

Rachel Law, rachel.law@csiro.au

---

## Author Response (AR2)

**Author's Response**

Dear Dr. Rachel Law,

Thank you very much for your comments and suggestions. We appreciate it very much. The answers are written in blue. All the suggestions have been included. Besides, Figure 4 has been updated due to mistaken R plotting codes regarding Mauna Loa weekly periodicity but results remain the same. And Figure 6 has been updated with a new dimension of width and height.

P3, line 8: replace 'combing' with 'combining'
Done.

P4, line 25: perhaps replace 'against' with 'to prevent'
Done.

P4, line 25: Suggest replace the sentence starting 'The inlet …' with 'The inlet was mounted on a small mast (approximately 4 m high) on the top of the laboratory building …'
Done.

P5, line 3: Replace 'the mast ends' with 'the mast height is'
Done.

P5, line 13-14: Perhaps include the sentence from your reply to reviewer 2 (4/5-6): 'For continuous quality assurance the GC was checked daily for flows, retention times, gas pressures, and the structure of chromatograms.' somewhere in this paragraph.
Thank you for your suggestion. We have included this sentence in this paragraph.

P5, line 13: Replace 'pollution list' with 'logged list of local pollution from working activities in the Research Station.'
Done.

P5, line 18-19: Suggest replacing 'a quasi-continuously measured second target' with 'a second target (measured approximately 25 times per day)'
Done.

P6, line 2-3: You mention the different calibration scales here, but the only information about which scale is used for which time period is in the supplementary material (Table S1). I think it might be useful to move this table back into the manuscript, perhaps with a sentence at the end of section 2.2 such as 'The different instruments and calibration scales used at each location are summarised in Table 1.' For ZSF, perhaps you should be more explicit about the HP not being available in 2012-2013, and also that the CDRS has run for longer than 2012-2013, but those are the only years used in this study. This information could be added as table footnotes.
Thank you for your suggestion. We have put the table from the supplement back to the manuscript, and label it as "Table 1". The sentence has been added at the end of Sect 2.2. All the corresponding changes regarding "Table 1" have been made throughout the manuscript.

Regarding measurements at ZSF, the following sentence is added in the table description,

"…At ZSF, $CO_2$ data from GC measurements were not available from 2012 to 2013 due to an instrumental failure, thus data from CRDS measurements were used in these two years for this study. However, CRDS measurements were performed in parallel from the same air inlet since 2011…"

P6, line 7: Add '(Table 1)' after 'in the standards', assuming the instrument table is moved back to the main manuscript, otherwise refer to Table S1.
Done.

P6, line 13-15: Suggest replace sentence starting 'Due to ..' with 'Since we have insufficient information to determine a physically derived correction to the ZPT $CO_2$ data, an offset adjustment was made for further analyses based on the offsets in data computed in the overlapping years.'
Done.

P7, line 17: Delete 'results' before 'represent'.
Done.

P7, line 30: Suggest adding '(without ADVS selection)' after 'are calculated'
Done.

P11, line 21: Check table numbering if move table 1 back from supplement.
Done.

P12, line 19 and 20: To be consistent, you might want to change 'amplitude' to 'difference' in these sentences.
Thank you for your suggestion. We decided to change all the "difference" regarding peak-to-trough and seasonal terms into "amplitude", in order to differ from the "difference" used in other places (such as, weekday to weekend difference, difference between values, etc.).

P12, last paragraph: Reviewer 1 suggests that one factor in differences in seasonal amplitude between Mount Zugspitze locations is the different measurement time periods and the possibility that seasonal amplitude has grown over time. This is probably worth noting in this paragraph, alongside the discussion of different air-mass transport to the different locations.
Thank you for your suggestion. We added at the end of the paragraph (as well as in the reference list),
"…On the other hand, these differences in the seasonal amplitudes (even though not significant at the 95% confidence interval) might be influenced by a potential trend in the seasonal amplitude over time. Such increasing trends of the seasonal $CO_2$ amplitudes (i.e., +0.32 % $yr^{-1}$ at Mauna Loa, and +0.60 % $yr^{-1}$ at Barrow, Alaska) were studied in Graven et al. (2013), indicating an enhanced interaction between the biosphere and atmospheric $CO_2$ across the Northern

Hemisphere…"

P13, line 13: Suggest adding 'original' before 'validated' and 'only' before '4634'.
Done.

P16, line 2: Add 'composite' before 'CO2 measurement record'
Done.

Figure 2 caption: For panel (a) the caption needs to state that grey and black are used for the unselected and selected results.
Done.

Figure 3: Please consider whether this figure would be easier to understand if the percentages were calculated relative to the number of data points available for that hour, rather than to the total number of data. This would presumably give percentages that would average (rather than sum) to the number quoted in Sec 3.1.
Done. Figure 3 has been updated, with the percentage was calculated as the number of ADVS-selected data per hour to the total number of $CO_2$ data for that hour.

Supplement, p2, line 16: Should be Table S1 not Table S2 (or Table 1) if this is moved into the paper as suggested above.
Done.

Supplement, p7, line 4: If move table, replace 'Table S1' with 'Table 1'
Done.

Supplement, Figure S7: You seem to have some odd behaviour at the start of the ZPT_ADVS timeseries with very low $CO_2$ concentrations. How has this occurred and will it have impacted any of your analysis?
Thank you for your comment. The irregular behavior of very low $CO_2$ concentrations in the STL decomposition plot was due to the greatly missing data at the beginning of the ZPT data set (1981 and 1982). It was only shown as STL decomposition output in the supplement. After ADVS data selection and STL decomposition, we made the first two years of ZPT data set as NA for all further analyses.
We compared it with the output when decomposing the ZPT time series directly starting from 1983. There were no significant differences in the results. Most main results were identical.
The reason has been stated in Sect 2.6 (STL decomposition) and we have rephrased it as,

[revised manuscript text omitted]

1982; Kirk Thoning, personal communication, August 1, 2018). Since we have insufficient information to determine a physically derived correction to the ZPT $CO_2$ data, an offset adjustment 
[revised manuscript text omitted]

| ID | Time period | Instrument (Analytical method) | Scale | Calibration gas |
|----|-------------|-------------------------------|-------|-----------------|
| ZPT | 1981–1997 | 1981–1984: Hartmann & Braun URAS 2 (NDIR) | WMO X74 scale | $CO_2$ in $N_2$ |
| | | 1985–1988: Hartmann & Braun URAS 2T (NDIR) | | |
| | | 1989–1997: Hartmann & Braun URAS 3G (NDIR) | | |
| ZUG | 1995–2001 | Hartmann & Braun URAS 3G (NDIR) | WMO X85 scale | $CO_2$ in natural air |
| ZSF | 2001–2016 | 2001–2016: Hewlett Packard Modified HP 6890 Chem. station (GC) | WMO X2007 scale | $CO_2$ in natural air |
| | | 2012–2013: Picarro EnviroSense 3000i (CRDS) | | |
| WNK | 1981–1996 | Hartmann & Braun URAS 2T (NDIR) | WMO X74 scale | $CO_2$ in $N_2$ |

**Table 12: Mean annual $CO_2$ growth rates in ppm yr$^{-1}$ at the 0.95 confidence interval based on three time blocks for all measurement sites / locations studied (SSL – Schauinsland; WNK – Mount Wank; ZPT – pedestrian tunnel at Mount Zugspitze; ZUG – Zugspitze summit; ZSF – Zugspitze Schneefernerhaus; MLO – Mauna Loa; WDCGG and NOAA – global means). ADVS means the data were selected by ADVS method. This comparison refers to data from all years including the corresponding time period for all stations. Measurement sites or locations where data are not available for calculating the corresponding time blocks are shown as "–".**

| Time block | SSL | WNK | ZPT | ZPT_ADVS | ZUG | ZUG_ADVS | ZSF | ZSF_ADVS | MLO | MLO_ADVS | WDCGG | NOAA |
|---|---|---|---|---|---|---|---|---|---|---|---|---|
| **1981–1994** | 1.5 ± 0.5 | 1.4 ± 1.1 | 1.5 ± 0.8 | 1.5 ± 1.4 | – | – | – | – | 1.4 ± 0.3 | 1.4 ± 0.3 | 1.4 ± 0.4 | 1.4 ± 0.3 |
| **1995–2001** | 1.7 ± 1.1 | – | – | – | 1.3 ± 0.8 | 1.5 ± 0.5 | – | – | 1.8 ± 0.5 | 1.8 ± 0.5 | 1.8 ± 0.4 | 1.7 ± 0.5 |
| **2002–2016** | 2.2 ± 0.7 | – | – | – | – | – | 2.2 ± 0.4 | 2.2 ± 0.4 | 2.2 ± 0.2 | 2.2 ± 0.2 | 2.2 ± 0.2 | 2.2 ± 0.2 |

[Figure]

**Figure 1: (a) Map showing the study area (GAP – Garmisch-Partenkirchen; WNK – Mount Wank; ZPT – pedestrian tunnel at Mount Zugspitze; ZUG – Zugspitze summit; ZSF – Zugspitze Schneefernerhaus). (b) A photograph showing the locations (ZPT, ZSF, and ZUG) at Mount Zugspitze where atmospheric $CO_2$ measurements were performed.**

[Figure]

Figure 2: a) Time series plot of 30-min averaged $CO_2$ concentrations measured at Mount Zugspitze (ZPT, ZUG, and ZSF) and Wank (WNK), and hourly averaged $CO_2$ concentrations measured at Schauinsland (SSL) and Mauna Loa (MLO) with  selection. Grey and black colors are used for the unselected and selected results. b) Detrended mean diurnal cycles with starting time windows (in grey) for ADVS data selection.

[Figure]

**Figure 3: Frequency of the percentages of the number of ADVS-selected CO$_2$ data for each hour (0 to 23) in the total number of CO$_2$ data . **

[Figure]

**Figure 4: a)** Mean MSR $CO_2$ values at Mount Zugspitze and MLO as a function of the weekday. Mean MSR values are adjusted such that they sum to 0. **b)** Detrended mean $CO_2$ diurnal cycles at ZSF by weekday from 2002 to 2016. Uncertainties at a 95% confidence interval are shown by the shaded areas.

[Figure]

**Figure 5: Mean diurnal plots at ZSF during 2016 by weekday for a) CO, b) NO, and c) the standardized daily passenger number at the Zugspitzplatt and Zugspitze summit combined.**

[Figure]

**Figure 6: Mean CO₂ seasonal cycles from the STL seasonal component at each measurement site or location. Uncertainties at a 95% confidence interval are shown by the shaded areas with corresponding color.**

[Figure]

**Figure 7:** a) Annual ADVS-selected percentages. b) Annual $CO_2$ growth rates and global means from the NOAA and the WDCGG. The calculated growth rates are shown at the beginning of the year. Since the time period starts in 1981, the values of growth rates start in 1982. WDCGG data is only available starting 1984. c) Annual $CO_2$ seasonal amplitudes.